# From Centerlines to Hemodynamics: Anisotropic RBF Decoders for Coronary Arteries

## Abstract

Accurate and rapid estimation of hemodynamic metrics, such as pressure and wall shear stress (WSS), is important for assessing the severity of Coronary Artery Disease (CAD). Existing approaches, including invasive Fractional Flow Reserve (FFR) measurements and computationally expensive Computational Fluid Dynamics (CFD) simulations, face challenges in invasiveness, cost, and speed. We present a learned surrogate for fast prediction of CFD-simulated coronary hemodynamics from vessel centerline geometry. The model encodes 1D vessel centerlines together with inlet flow rate using a transformer-based encoder, and predicts continuous wall-based fields via an anisotropic Radial Basis Function (RBF) decoder aligned with vessel morphology. To support training and evaluation, we introduce two datasets with paired steady-state OpenFOAM simulations: (i) a synthetic benchmark of 4,200 single-vessel geometries with controlled anatomical variations, and (ii) a multi-vessel dataset derived from ImageCAS including 4,800 cases spanning both right and left coronary arteries, generated by randomly introducing stenoses and varying physiologically plausible flow rates. Across both datasets, our method achieves lower pressure and WSS errors than strong neural-operator baselines (GNOT, Transolver, and ONO) at a fraction of the computational cost of CFD. On the multi-vessel dataset, using 1,024 anisotropic RBF centers our model reduces the mean relative $\ell_2$ error by 52% compared to the best neural-operator baseline, while at 128 centers it requires 13.8× fewer FLOPs than GNOT and still outperforms all neural-operator baselines. The single-vessel dataset is publicly available.[1]

## 1 Introduction

Cardiovascular disease remains the leading cause of mortality worldwide, with coronary artery disease (CAD) accounting for a large fraction of deaths Virani & et al. (2021). Clinical decision-making in CAD depends on determining whether a coronary stenosis significantly impairs blood flow. The gold standard for this assessment is fractional flow reserve (FFR), defined as the ratio of distal to proximal coronary pressure during pharmacologically induced hyperemia Tonino et al. (2009). Despite its diagnostic value, routine FFR measurement is limited by the need for catheterization, pressure-wire instrumentation, and pharmacological intervention Pijls et al. (1996).

Patient-specific computational fluid dynamics (CFD) offers a non-invasive alternative by solving the incompressible Navier-Stokes equations on reconstructed vascular domains, accurately recovering pressure and wall shear stress (WSS) fields and enabling FFR estimation from imaging Min et al. (2015). However, generating high-quality meshes and solving three-dimensional flow fields typically requires hours of computation per case Sankaran et al. (2012). This precludes use in settings that demand rapid turnaround: point-of-care FFR estimation during a single imaging session, interactive exploration of interventional scenarios (e.g., virtual stent placement), and cohort-scale hemodynamic phenotyping where thousands of cases must be processed Corral-Acero et al. (2020). Learned surrogates operating at millisecond-scale inference could address these needs.

---

[1] Dataset URL withheld for anonymous review

Machine learning surrogates have been explored to reduce this cost. Physics-informed neural networks (PINNs) Raissi et al. (2019); Bafghi & Raissi (2023) embed governing equations into training but require retraining per geometry and struggle with multi-scale optimization difficulties Wang et al. (2020; 2023). Neural operators learn mappings between function spaces Kovachki et al. (2023) and avoid this limitation: DeepONet Lu et al. (2019) uses branch-trunk decomposition; FNO Li et al. (2020), Geo-FNO Li et al. (2022), and GINO Li et al. (2023) extend spectral learning to irregular geometries; and transformer-based operators including GNOT Hao et al. (2023), Transolver Wu et al. (2024), and ONO Xiao et al. (2023) further improve expressiveness through attention mechanisms. Geometric deep learning methods have also been applied directly to coronary artery surface meshes Nannini et al. (2025); Suk et al. (2024b;a); Kuang et al. (2024); Rygiel et al. (2023).

Despite this progress, a structural gap remains. Coronary arteries are tubular domains naturally parameterized by a one-dimensional centerline with an associated radius field, while clinically relevant outputs are defined on the vessel wall. Existing neural operator architectures operate on volumetric or surface discretizations and do not exploit this intrinsic low-dimensional structure, incurring unnecessary computational cost and failing to provide a unified framework for geometry-aware encoding with continuous, mesh-independent field reconstruction.

We address this gap by learning the nonlinear solution operator

$$\mathcal{G} : (\Gamma, \, r(s), \, q_{\text{in}}) \, \mapsto \, \big(p(\mathbf{x}), \, \tau_w(\mathbf{x})\big),$$

where $\Gamma$ is the vessel centerline, $r(s)$ is the radius field, $q_{\text{in}}$ is the inlet flow rate, and $(p, \tau_w)$ are spatially continuous pressure and WSS fields at arbitrary wall locations $\mathbf{x}$. The model is trained and evaluated entirely on synthetic CFD data; it serves as a fast surrogate for the simulation pipeline rather than a replacement for clinical measurement. Our Transformer-Anisotropic RBF Network encodes the 1D centerline via Fourier positional embeddings and a transformer encoder, conditions on inlet flow rate through Feature-wise Linear Modulation (FiLM), and decodes continuous fields as a weighted superposition of anisotropic Gaussian RBF kernels centered along the vessel. Each kernel carries a learned full-precision matrix, allowing orientation and scale to adapt to local vessel geometry. Evaluation at arbitrary query points is achieved by kernel aggregation at near-constant cost regardless of surface resolution.

**Contributions**

- **Geometry-aware operator formulation.** A reduced-order representation that exploits the intrinsic 1D centerline structure of tubular geometries for efficient hemodynamic surrogate modeling.

- **Mesh-free continuous field reconstruction.** An anisotropic RBF decoder enabling continuous pressure and WSS evaluation independent of surface discretization, with inference cost nearly invariant to query-point count.

- **Large-scale synthetic benchmark.** Two paired CFD datasets (4,200 single-vessel and 4,800 multi-vessel geometries) supporting reproducible evaluation; the single-vessel set is publicly released.

- **Improved accuracy/efficiency trade-off.** Lower $\ell_2$ errors than GNOT, Transolver, and ONO on both datasets, with up to 13.8× fewer FLOPs at matched RBF count.

The remainder of the paper is organized as follows. Section 2 reviews related work. Section 3 describes the methodology. Section 4 presents the datasets. Section 5 reports experimental results. Section 6 discusses limitations and future directions.

## 2 Related Work

**Computational hemodynamics and reduced-order modeling.** Patient-specific CFD remains the reference standard for coronary hemodynamics, solving the incompressible Navier-Stokes equations on anatomically reconstructed domains to recover pressure, WSS, and FFR Min et al. (2015); Sankaran et al. (2012). Its computational expense, however, limits use in time-sensitive and large-scale settings.

Classical reduced-order modeling (ROM) addresses this through projection-based methods such as proper orthogonal decomposition (POD) and reduced basis approaches, which construct low-dimensional subspaces from high-fidelity simulations. While these provide substantial savings with physical interpretability, they rely on linear approximations and often require intrusive access to the governing equations. One-dimensional and lumped-parameter models offer further efficiency along vessel centerlines but cannot capture three-dimensional effects such as secondary flows and localized WSS distributions. Hybrid data-driven ROM approaches have been proposed to reconstruct pressure and WSS from limited simulation data Morgan et al. (2023), and graph neural networks have been applied to learn reduced-order cardiovascular flow models Pegolotti et al. (2023), but generalization across complex geometries remains limited.

**Machine learning for hemodynamic prediction.** PINNs Raissi et al. (2019); Bafghi & Raissi (2023); Kissas et al. (2020) embed governing equations as training regularizers but solve individual PDE instances rather than amortized operators, and optimization for multi-scale flows is sensitive to loss balancing Wang et al. (2020). Geometric deep learning methods applied to coronary artery surface meshes Nannini et al. (2025); Suk et al. (2024b;a) and physics-informed self-supervised approaches for hemodynamic digital twins Kuang et al. (2024) have shown promise, but their cost scales with the number of discretization points. CenterlinePointNet++ Rygiel et al. (2023) also leverages centerline and surface point cloud representations for coronary pressure drop and vFFR estimation, but predicts only a scalar per vessel rather than spatially continuous wall fields. We adapt this architecture for spatially resolved field prediction and include it as a domain-specific baseline (Section 5). In a concurrent line of work, physics-informed approaches have also been applied to estimate coronary flow reserve directly from angiography Thakur et al. (2026).

**Neural operators.** Neural operators learn resolution-independent mappings between function spaces Kovachki et al. (2023), avoiding the per-instance retraining of PINNs. DeepONet Lu et al. (2019) uses branch-trunk decomposition; FNO Li et al. (2020), Geo-FNO Li et al. (2022), and GINO Li et al. (2023) extend spectral convolution to irregular geometries via learned coordinate transforms or graph representations; and transformer-based operators GNOT Hao et al. (2023), Transolver Wu et al. (2024) (recently extended to million-scale meshes as Transolver++ Luo et al. (2025)), and ONO Xiao et al. (2023) add long-range expressiveness through attention. Universal Physics Transformers Alkin et al. (2024) provide a general encoder-approximator-decoder framework for scaling neural operators across discretization types. Despite these advances, all these families operate on volumetric or surface discretizations and do not exploit the intrinsic one-dimensional structure of tubular vascular geometries.

**Mesh-free and kernel-based representations.** RBF approximations have a long history in computational mechanics for mesh-free interpolation, constructing solutions as weighted superpositions of kernel functions centered at selected locations Fasshauer (2007). Anisotropic extensions using ellipsoidal support regions have been explored for partition-of-unity interpolation Cavoretto et al. (2018), and recent work has revisited learned anisotropic RBF kernels in regression settings Gerber & Lloyd (2026). Classical RBF methods, however, use fixed kernels and do not adapt to parameter-dependent solution behavior in an operator-learning context.

**Gap.** No existing approach combines geometry-aware encoding of the 1D centerline structure with continuous, mesh-independent field reconstruction in a unified operator framework. This is the gap our method addresses.

## 3 Methodology

Our proposed Transformer-Anisotropic RBF Network predicts continuous pressure and wall shear stress (WSS) fields on coronary artery walls from vessel geometry and inlet flow rate. The framework combines a transformer-based encoder for geometry-aware feature extraction with an anisotropic Radial Basis Function (RBF) decoder for continuous field reconstruction (Fig. 1).

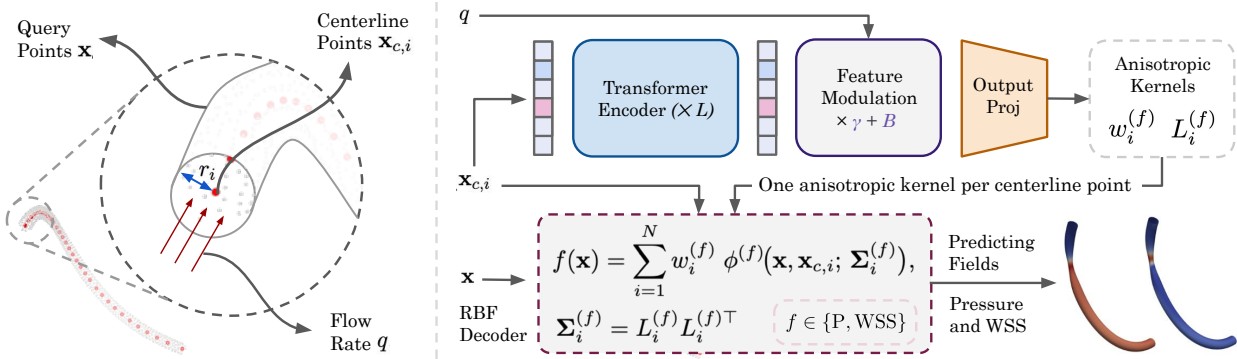

Figure 1: Architecture of the Transformer-Anisotropic RBF Network. The transformer encoder processes centerline geometry and flow rate, producing parameters for anisotropic RBF kernels aligned with vessel morphology to reconstruct continuous pressure and WSS fields.

## 3.1 Problem Formulation

Given a vessel centerline $\mathbf{C} = \{(\mathbf{x}_i, r_i)\}_{i=1}^M$ with coordinates $\mathbf{x}_i \in \mathbb{R}^3$, local radii $r_i \in \mathbb{R}^+$, and inlet flow rate $q \in \mathbb{R}$, we learn the operator

$$\mathcal{G} : (\mathbf{C}, q) \mapsto (\mathrm{P}(\mathbf{x}), \mathrm{WSS}(\mathbf{x})),$$

predicting pressure and WSS magnitude at arbitrary query points $\mathbf{x}$ on the vessel wall. Figure 2 illustrates this mapping for single- and multi-vessel anatomies.

## 3.2 Architecture Overview

The model has two main components: (1) a transformer encoder that processes centerline geometry and flow rate, and (2) an anisotropic RBF decoder that reconstructs continuous pressure and WSS fields on the wall surface.

**Transformer Encoder with 4D Fourier.** Each centerline point $\mathbf{c}_i = (x_i, y_i, z_i, r_i)$ is encoded with sinusoidal positional embeddings Tancik et al. (2020) applied independently to each of the four input coordinates at $K$ exponentially spaced frequencies:

$$\mathbf{e}(\mathbf{c}) = \big[ \sin(\omega_k\, c_j),\ \cos(\omega_k\, c_j) \big]_{\substack{j=1,\dots,4 \\ k=0,\dots,K-1}} \in \mathbb{R}^{8K}, \quad \omega_k = \pi \cdot 2^k. \tag{1}$$

The embedding is concatenated with the raw coordinates and linearly projected to the model dimension $d$:

$$\mathbf{h}_i^{(0)} = \mathbf{W}\big[\mathbf{e}(\mathbf{c}_i)\,;\,\mathbf{c}_i\big] + \mathbf{b}, \quad \mathbf{W} \in \mathbb{R}^{d \times (8K+4)}. \tag{2}$$

Learned positional embeddings are added to the token sequence, which is then processed by a stack of transformer encoder layers Vaswani et al. (2017) to produce features $\{\mathbf{h}_i\}_{i=1}^M$.

**Flow Rate Conditioning.** After the transformer encoder, the inlet flow rate $q$ modulates the encoded features via Feature-wise Linear Modulation (FiLM) Pérez et al. (2018). A two-layer MLP maps the scalar flow rate to scale and shift vectors:

$$\boldsymbol{\gamma}, \boldsymbol{\beta} = \mathrm{MLP}(q), \quad \boldsymbol{\gamma}, \boldsymbol{\beta} \in \mathbb{R}^d, \tag{3}$$

$$\tilde{\mathbf{h}}_i = \boldsymbol{\gamma} \odot \mathbf{h}_i + \boldsymbol{\beta}, \tag{4}$$

where $\odot$ denotes element-wise multiplication broadcast over all $M$ centerline tokens.

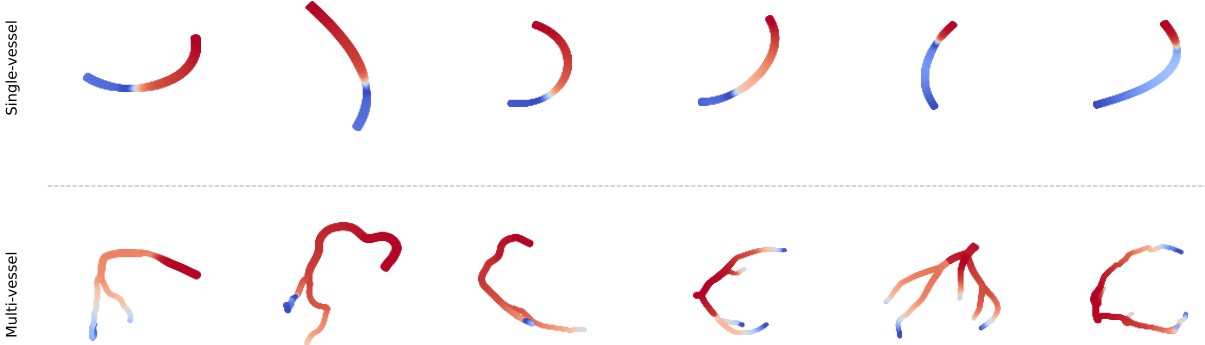

Figure 2: Diverse coronary artery geometries from the single-vessel (top) and multi-vessel (bottom) datasets, colored by pressure. Multi-vessel cases include right coronary arteries (RCA, first three) and left coronary artery (LCA, last three) trees with varying branching complexity.

**Anisotropic RBF Decoder.** A linear layer maps each modulated feature $\tilde{\mathbf{h}}_i$ to 14 parameters per centerline point: for each field $f \in \{\mathrm{P}, \mathrm{WSS}\}$, a scalar weight $w_i^{(f)}$ and six entries of a lower-triangular Cholesky factor $\mathbf{L}_i^{(f)} \in \mathbb{R}^{3\times3}$ (with positive diagonal enforced via squaring). The Cholesky factor defines a positive-definite precision matrix $\mathbf{\Sigma}_i^{(f)\,-1} = \mathbf{L}_i^{(f)} \mathbf{L}_i^{(f)\top}$, yielding an anisotropic Gaussian kernel centered at centerline coordinate $\mathbf{x}_{c,i}$:

$$\phi^{(f)}(\mathbf{x}, \mathbf{x}_{c,i}) = \exp\left(-(\mathbf{x} - \mathbf{x}_{c,i})^\top \mathbf{\Sigma}_i^{(f)\,-1} (\mathbf{x} - \mathbf{x}_{c,i})\right). \tag{5}$$

Given an arbitrary query point $\mathbf{x}$ on the vessel wall, the predicted fields are obtained as weighted sums over all $M$ kernels, one per centerline point:

$$\mathrm{P}(\mathbf{x}) = \sum_{i=1}^{M} w_i^{(\mathrm{P})}\, \phi^{(\mathrm{P})}(\mathbf{x}, \mathbf{x}_{c,i}), \quad \mathrm{WSS}(\mathbf{x}) = \sum_{i=1}^{M} w_i^{(\mathrm{WSS})}\, \phi^{(\mathrm{WSS})}(\mathbf{x}, \mathbf{x}_{c,i}). \tag{6}$$

## 4 Dataset

We use two datasets in this work: a synthetic single-vessel dataset and a multi-vessel coronary artery dataset. Both datasets are generated with controlled anatomical variations and paired with steady-state flow simulations. Figure 2 shows representative geometries from both datasets.

### 4.1 Single-Vessel Dataset

To construct the single-vessel dataset, we generated synthetic coronary vessel geometries with controlled anatomical variations. The vessel length ranged from $40\,\mathrm{mm}$ to $70\,\mathrm{mm}$, tapering ratios (defined as the ratio of outlet to inlet radius) varied between 0.6 and 0.8, and stenosis severity ranged from 30% to 70%. These parameters were used to generate vessel centerline representations consisting of coordinates and radius values, denoted as $(x, y, z, r)$.

Using these centerlines, we constructed 3D surface and volume meshes and performed steady-state flow simulations using OpenFOAM Jasak (2009). Physiologically relevant inlet pressures and flow rates were prescribed for each case. In total, 4,200 single-vessel geometries were generated along with their corresponding flow fields. The dataset was partitioned into 3,600 training, 400 validation, and 200 test cases, with fixed splits held constant across all experiments to ensure reproducibility.

Table 1: Performance on synthetic single-vessel geometries (3,600 train / 400 val / 200 test). Test relative $\ell_2$ error (mean±std over 3 seeds, ↓). Validation results are in Appendix Table 12.

| Method | Pressure | WSS | Mean |
|---|---|---|---|
| Low-Fidelity | 0.635 | 0.488 | 0.562 |
| GNOT Hao et al. (2023) | 0.133±0.008 | 0.164±0.012 | 0.149±0.010 |
| Transolver Wu et al. (2024) | 0.188±0.046 | 0.246±0.069 | 0.217±0.057 |
| ONO Xiao et al. (2023) | 0.148±0.046 | 0.143±0.033 | 0.145±0.039 |
| Ours (64 RBFs) | 0.126±0.011 | 0.163±0.010 | 0.145±0.010 |
| Ours (128 RBFs) | 0.101±0.004 | 0.134±0.010 | 0.117±0.007 |
| Ours (256 RBFs) | 0.124±0.003 | 0.154±0.003 | 0.139±0.001 |
| Ours (512 RBFs) | **0.099±0.012** | **0.133±0.012** | **0.116±0.012** |

### 4.2 Multi-Vessel Dataset

In addition to the single-vessel data, we constructed a multi-vessel coronary artery dataset based on anatomies from the ImageCAS dataset Zeng et al. (2023), which contains coronary artery segmentations from 1,000 patients. We extract both right coronary artery (RCA) and left coronary artery (LCA) centerlines from 140 patients, yielding distinct vessel trees that vary in branching topology and curvature. For each anatomy, we generate multiple simulation cases by (i) introducing synthetic stenoses at random locations with varying severities, and (ii) assigning different physiologically plausible inlet pressures and volumetric flow rates. This produces up to 10 variants per base geometry, each with different hemodynamic boundary conditions.

For each variant, 3D meshes were generated and steady-state flow simulations were performed using the same OpenFOAM pipeline as in the single-vessel dataset (simpleFoam solver, Newtonian fluid, no-slip walls; see Appendix A.1 for simulation details and Appendix A.2 for preprocessing). The final multi-vessel dataset consists of 4,000 training, 400 validation, and 400 test cases, with fixed splits across all experiments. Across all splits, the dataset contains 3,486 RCA and 1,314 LCA cases. To evaluate generalization to unseen anatomies, we additionally construct a patient-disjoint re-split (3,989 / 415 / 396) in which no patient's anatomy appears in more than one of {train, val, test}; details are provided in Appendix A.13.

## 5 Experiments

We evaluate our method on both (i) synthetic single-vessel geometries and (ii) realistic multi-vessel coronary anatomies (RCA/LCA) to assess accuracy, robustness, and computational efficiency. Across all experiments, we compare against neural operator baselines, including GNOT Hao et al. (2023), Transolver Wu et al. (2024), and ONO Xiao et al. (2023), as well as the domain-specific CenterlinePointNet++ Rygiel et al. (2023) on multi-vessel data. All models are trained with comparable parameter counts (∼2.9M) to predict both pressure and WSS fields (see Appendix A.4 for architecture details). All models were trained for 5,000 epochs on single-vessel data and 3,000 epochs on multi-vessel data. All results are reported as mean±standard deviation over three random seeds from the checkpoint with the lowest validation loss (see Appendix A.3). Evaluation metrics are defined in Appendix A.6.

### 5.1 Predictive Accuracy

**Single-vessel results.** Table 1 reports relative $\ell_2$ errors on the test split (200 cases) for our model at 64, 128, 256, and 512 RBFs alongside the baselines. Our model obtains the lowest mean error at 512 RBFs (test mean 0.116±0.012), with particularly strong pressure predictions. Among the baselines, ONO and GNOT are competitive, while Transolver shows high variance across seeds and lags behind. Validation results are in Appendix Table 12.

**Multi-vessel results.** We extend the evaluation to a multi-vessel coronary artery dataset derived from ImageCAS Zeng et al. (2023), containing both RCA and LCA anatomies. Because multi-vessel geometries

Table 2: Performance on multi-vessel RCA/LCA geometries (4,000 train / 400 val / 400 test). Test relative $\ell_2$ error (mean±std over 3 seeds, ↓). [†]Adapted from original (see Appendix A.4). Validation results are in Appendix Table 13.

| Method | Pressure | WSS | Mean |
|---|---|---|---|
| Low-Fidelity | 0.852 | 0.683 | 0.768 |
| GNOT Hao et al. (2023) | 0.588±0.003 | 0.717±0.008 | 0.653±0.005 |
| Transolver Wu et al. (2024) | 0.585±0.020 | 0.699±0.007 | 0.642±0.011 |
| ONO Xiao et al. (2023) | 0.592±0.002 | 0.717±0.008 | 0.654±0.003 |
| CenterlinePointNet++[†] Rygiel et al. (2023) | 0.472±0.002 | **0.315±0.002** | 0.393±0.000 |
| Ours (128 RBFs) | 0.429±0.015 | 0.459±0.010 | 0.444±0.013 |
| Ours (256 RBFs) | 0.397±0.011 | 0.401±0.008 | 0.399±0.006 |
| Ours (512 RBFs) | 0.327±0.013 | 0.373±0.004 | 0.350±0.005 |
| Ours (1024 RBFs) | **0.254±0.015** | 0.355±0.005 | **0.304±0.010** |

vary widely in length and branching complexity, we study the effect of the number of RBF bases on this dataset. Table 2 reports results for our model at 128, 256, 512, and 1,024 RBFs alongside the baselines. All generic neural-operator baselines struggle on this more complex dataset (test mean $\ell_2 > 0.64$), while the 1D Poiseuille low-fidelity model fares even worse ($\ell_2 = 0.768$). The domain-specific CenterlinePointNet++, which explicitly leverages centerline geometry via hierarchical point cloud encoding, substantially outperforms the generic baselines, confirming the benefit of incorporating vascular topology. Nevertheless, our model outperforms CenterlinePointNet++ at 512 RBFs and above, with performance improving consistently as the number of bases increases. The best configuration (1,024 RBFs) reaches a test mean $\ell_2$ of 0.304±0.010, roughly half the best generic baseline error. A failure-case analysis (Appendix A.14) shows that the remaining errors concentrate on geometrically complex LCA anatomies and patients with unusually tight stenoses, representing fewer than 2% of cases. Figure 3 visualizes the pointwise absolute error on representative RCA and LCA test cases, confirming that our model produces substantially lower errors across both pressure and WSS fields. A qualitative comparison on single-vessel data is provided in Appendix A.10.

**Effect of RBF size.** Tables 1 and 2 show how the number of RBF bases affects accuracy. On single-vessel data, 512 RBFs achieves the best test performance (mean $\ell_2 = 0.116\pm0.012$), with a general trend of improvement from 64 to 512. The 256-RBF configuration is a slight exception (test mean 0.139 vs. 0.117 for 128 RBFs). On multi-vessel data, where vessel lengths and branching complexity vary more, accuracy improves consistently up to 1,024 bases (test mean $\ell_2 = 0.304\pm0.010$), suggesting that complex geometries benefit from higher spatial resolution in the RBF representation.

Per-case error distributions (Figure 4) confirm that our model exhibits substantially lower median error and tighter interquartile range than all baselines on multi-vessel data for both pressure and WSS. An analysis of circumferential WSS reconstruction is provided in Appendix A.11, and training-set error comparisons are in Appendix A.12.

**Patient-disjoint evaluation.** The multi-vessel dataset contains multiple stenosis and boundary-condition variants per patient anatomy. To verify that results are not inflated by anatomy leakage between splits, we construct a patient-disjoint re-split where no patient's anatomy appears in more than one of {train, val, test} (see Appendix A.13 for details). Table 3 reports test-set results under this harder evaluation.

Under the patient-disjoint split, all methods degrade as expected: unseen anatomies are harder than unseen boundary conditions on known geometries. The baselines degrade by $\sim10\%$ (test mean $0.65 \rightarrow 0.71$ to $0.72$). Our model achieves a test mean of 0.558±0.006, still outperforming all baselines by a wide margin (22% lower error than the best baseline). This confirms that the centerline-anchored architecture provides genuine generalization advantage even when the inductive bias of familiar geometry is removed.

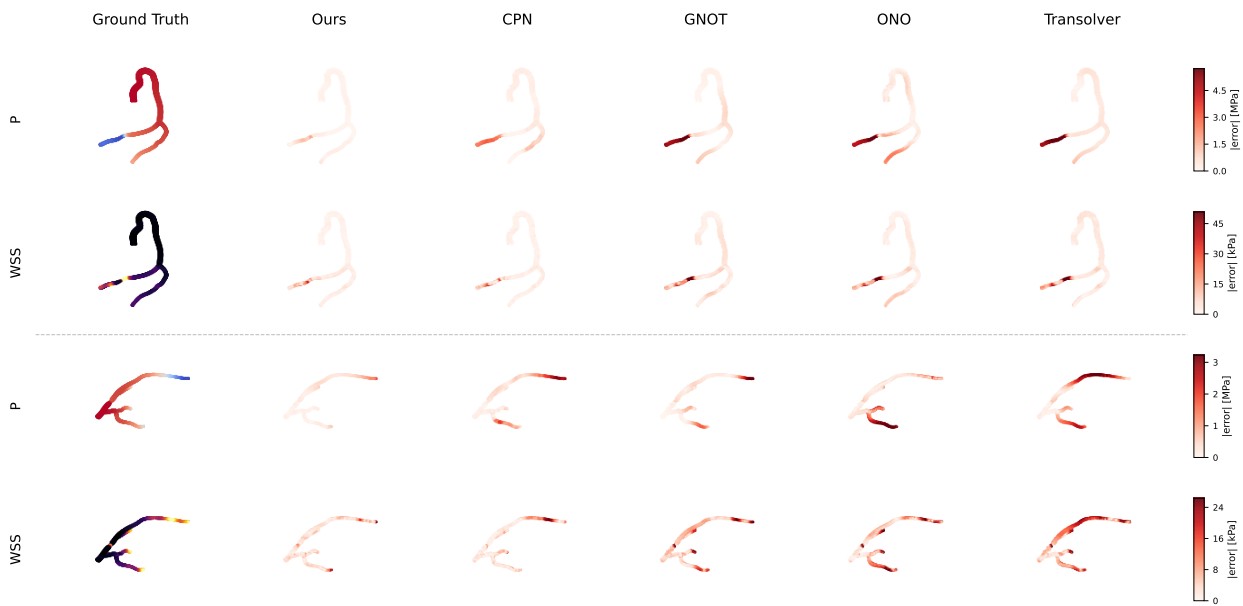

Figure 3: Pointwise absolute prediction error on two multi-vessel test geometries (top: RCA, bottom: LCA). The first column shows the ground-truth CFD solution; columns 2 to 6 display $|\hat{y} - y|$ for each model on a shared color scale. Our model (1024 RBFs) correctly captures the pressure drop and WSS elevation at stenosis locations.

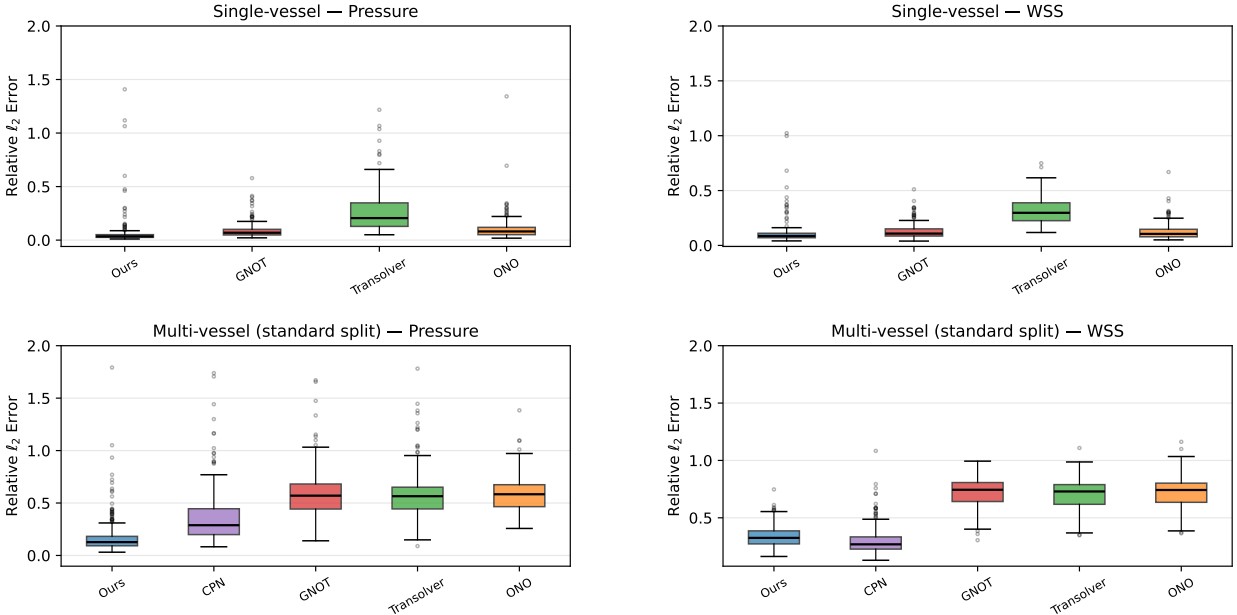

Figure 4: Per-case relative $\ell_2$ error distributions on the test set. Top row: single-vessel; bottom row: multi-vessel (standard split). Each box shows the median, interquartile range, and outliers across all test cases. Our model achieves lower and more consistent errors than all baselines on multi-vessel data.

**Stratified performance analysis.** To further characterize generalization, we stratify test-set errors on the standard multi-vessel split by flow rate regime (in-distribution vs. near-boundary, defined as above the 90th percentile of training flow rates), flow rate tercile, vessel type (RCA vs. LCA), and branching

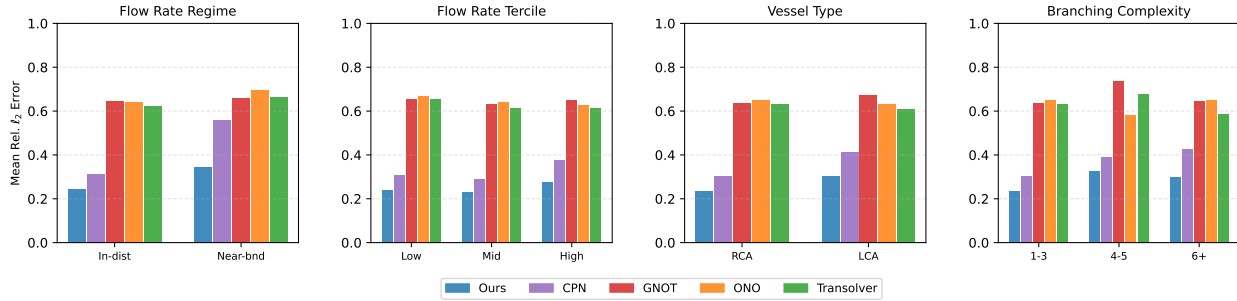

Figure 5: Stratified test-set performance on multi-vessel data (standard split). Columns show error grouped by (1) flow rate regime relative to training distribution, (2) flow rate tercile, (3) vessel type, and (4) branching complexity. Our model (blue) outperforms baselines across most conditions, with the gap narrowing on complex LCA geometries with many branches.

Table 3: Patient-disjoint multi-vessel evaluation (3,989 train / 415 val / 396 test). Test relative $\ell_2$ error (mean±std over 3 seeds) for pressure and WSS.

| Method | Pressure | WSS | Mean |
|---|---|---|---|
| GNOT Hao et al. (2023) | 0.685±0.006 | 0.754±0.005 | 0.720±0.005 |
| Transolver Wu et al. (2024) | 0.691±0.012 | 0.749±0.012 | 0.720±0.011 |
| ONO Xiao et al. (2023) | 0.680±0.011 | 0.741±0.011 | 0.710±0.010 |
| Ours (128 RBFs) | **0.533±0.012** | **0.584±0.005** | **0.558±0.006** |

complexity (Figure 5). Our model maintains a substantial advantage over all baselines across most conditions. CenterlinePointNet++ consistently outperforms the generic baselines in every subgroup, confirming the benefit of centerline-aware architectures. The gap between models narrows on complex LCA geometries with 6+ branches, where all methods degrade.

**Input noise robustness.** We evaluate inference-time sensitivity to imperfect inputs on the multi-vessel test set without retraining, applying Gaussian noise to surface coordinates, Gaussian noise to inlet flow rate, and random surface point dropout (Figure 6). Our model maintains the lowest absolute error across all noise levels. This robustness is partly architectural: our encoder processes centerline tokens (unperturbed), using surface points only as RBF query locations, whereas baselines and CenterlinePointNet++ process surface points directly and degrade more under coordinate noise and dropout (up to 3.9× for CenterlinePointNet++ at 50% dropout). Noise-aware training is left for future work.

**FFR evaluation.** We compare predicted fractional flow reserve (FFR) against ground truth for all methods on the test set (Figure 7). Our model obtains the lowest FFR MAE on both datasets and the highest correlation with ground truth. The low-fidelity 1D Poiseuille baseline is essentially uncorrelated with the ground truth on multi-vessel data. Detailed FFR metrics (MAE and classification accuracy) are reported in Appendix A.7.

## 5.2 Computational Efficiency

To analyze the relationship between computational cost and performance, we measure giga floating-point operations (GFLOPs) for a single forward pass under two controlled experiments: (i) fixing the number of RBFs and varying the number of query surface points, and (ii) fixing the query count and varying the number of RBFs. As shown in Figure 8, our method achieves better accuracy per FLOP than all baselines. At 128 RBF centers, our model requires only 0.53 GFLOPs, which is 13.8× fewer than GNOT (7.31), 19.6× fewer than ONO (10.37), 8.7× fewer than Transolver (4.59), and 6.3× fewer than CenterlinePointNet++ (3.32). At 64 centers, the cost drops further to just 0.24 GFLOPs. At 512 centers the cost rises to 3.12 GFLOPs

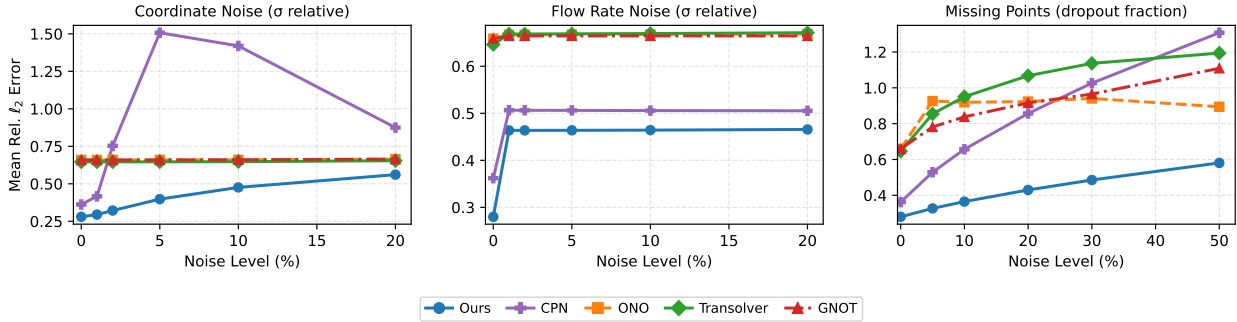

Figure 6: Input noise robustness on the multi-vessel test set (standard split). Columns show inference-time degradation under (left) Gaussian coordinate noise applied to surface mesh points, (middle) Gaussian noise on inlet flow rate, and (right) random surface point dropout. All models use checkpoints trained on clean data with comparable configurations. Curves report mean relative $\ell_2$ error averaged over pressure and WSS; each noise level is averaged over three random perturbation trials.

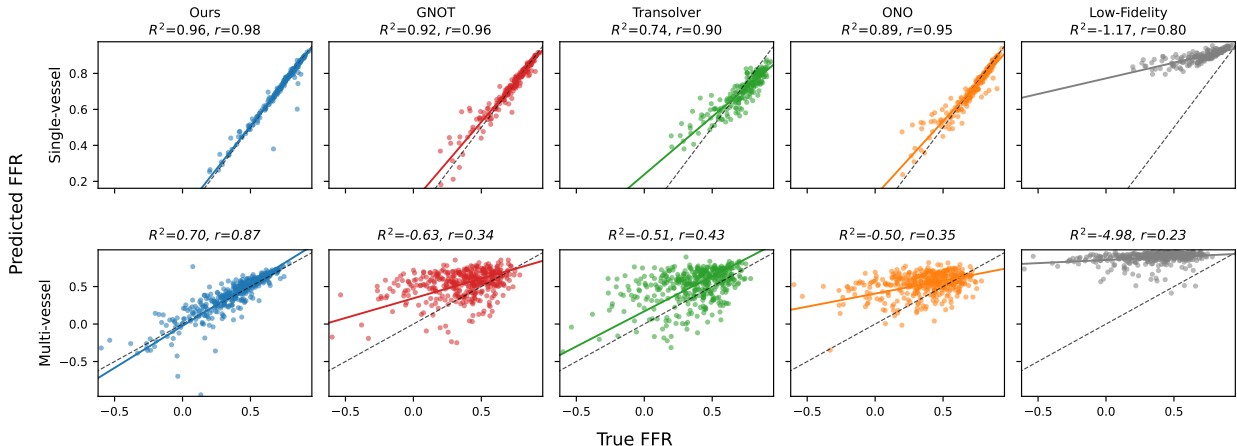

Figure 7: Predicted vs. true FFR on the test set. Top row: single-vessel; bottom row: multi-vessel. Columns show our model, three neural-operator baselines, and the 1D Poiseuille low-fidelity model. Our model obtains the highest $R^2$ and correlation on both datasets. The low-fidelity model systematically overestimates FFR (underestimates pressure drop), predicting most cases as non-significant, and is essentially uncorrelated with the ground truth on multi-vessel data.

but remains $2.3\times$ lower than GNOT and $3.3\times$ lower than ONO. Crucially, the shallow RBF decoder makes inference cost nearly invariant to the number of query points (0.51-0.53 GFLOPs from 256 to 2,048 points), whereas baselines scale steeply (up to $8\times$ increase over the same range for GNOT and Transolver). This is because the transformer encoder accounts for $96\,\text{to}\,98\%$ of total FLOPs across all configurations; the decoder contributes only $2\,\text{to}\,4\%$ and is the only component that scales with query count (Appendix A.9). Wall-clock timing benchmarks confirming these ratios are provided in Appendix A.8.

## 5.3 Ablation Study

We ablate four design choices on the single-vessel dataset using $M{=}128$ centerline points trained for 300 epochs. Starting from the default configuration (Table 4, row 1), each variant modifies exactly one component. Recall that the RBF decoder predicts, for each centerline point $i$ and each field $f \in \{\mathrm{P}, \mathrm{WSS}\}$, a weight $w_i^{(f)}$ and a lower-triangular Cholesky factor $\mathbf{L}_i^{(f)}$ whose diagonal must be positive. The four axes we vary are as follows.

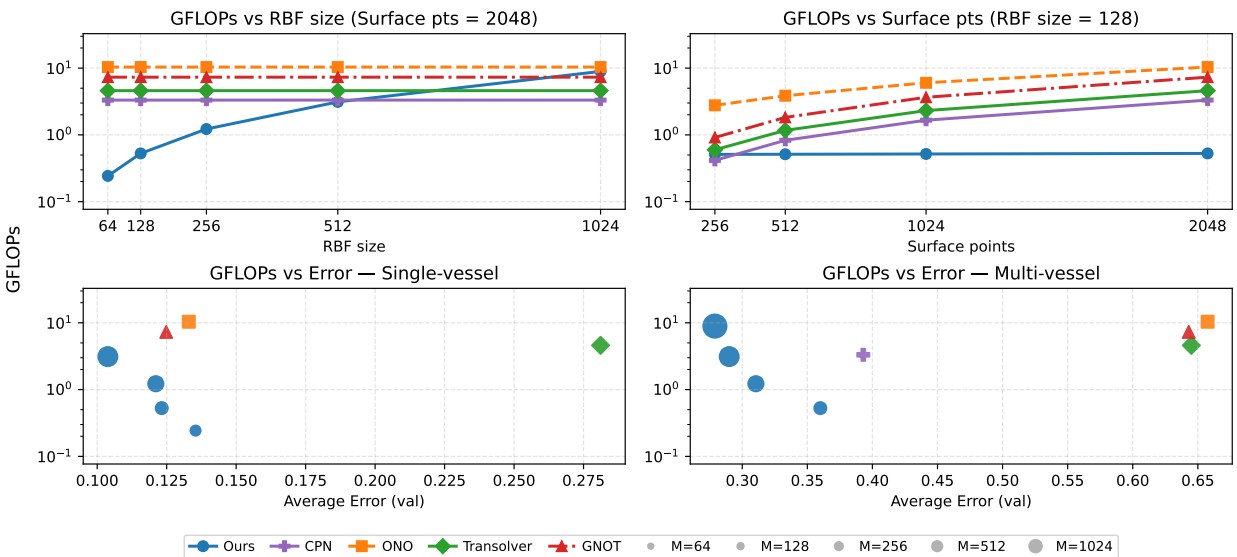

Figure 8: Computational cost vs. accuracy. (Top-left) GFLOPs increase with RBF count but remain comparable to baselines at 128 bases. (Top-right) Our model's cost is nearly invariant to the number of query points, unlike baselines that scale steeply. (Bottom) Our model achieves lower error at lower computational cost on both single-vessel (left) and multi-vessel (right) data.

Table 4: Design choice ablation on single-vessel data ($M$=128, 300 epochs). Each row modifies one component from the baseline. Metrics are relative $\ell_2$ error at the best validation epoch.

| Component | Variant | Pressure | WSS | Mean |
|---|---|---|---|---|
| *Baseline* (FiLM, squared, aniso, separate) | | 0.113 | **0.210** | **0.162** |
| Conditioning | CLS token | **0.112** | 0.224 | 0.168 |
| Positivity | Softplus | 0.137 | 0.338 | 0.237 |
| Kernel shape | Isotropic | 0.121 | 0.338 | 0.230 |
| Kernel sharing | Shared (P & WSS) | 0.122 | 0.253 | 0.187 |

**Conditioning mechanism.** FiLM Pérez et al. (2018) applies a learned affine modulation $\tilde{\mathbf{h}}_i = \boldsymbol{\gamma}(q) \odot \mathbf{h}_i + \boldsymbol{\beta}(q)$ uniformly to all tokens after the encoder. The CLS alternative prepends a flow-rate token $\mathbf{t}_q = \phi_q(q)$ to the sequence, letting the transformer mix it via self-attention, and discards it before decoding.

**Diagonal positivity.** The diagonal entries of $\mathbf{L}_i^{(f)}$ are mapped to $\mathbb{R}^+$ via either $g(x) = x^2 + \epsilon$ (squared) or $g(x) = \log(1 + e^x) + \epsilon$ (softplus).

**Kernel shape.** The full anisotropic kernel uses the precision matrix $\boldsymbol{\Sigma}_i^{-1} = \mathbf{L}_i \mathbf{L}_i^\top \in \mathbb{R}^{3 \times 3}$ (6 free parameters). The isotropic variant replaces it with a single scalar bandwidth $\sigma_i$:

$$\phi_{\text{iso}}(\mathbf{x}, \mathbf{x}_{c,i}) = \exp\left(-\sigma_i^2 \|\mathbf{x} - \mathbf{x}_{c,i}\|^2\right). \tag{7}$$

**Kernel sharing.** The default model learns independent kernel parameters for pressure and WSS ($2 \times 6$=12 covariance parameters per point). The shared variant uses a single set of 6 parameters for both fields.

The baseline configuration achieves the lowest mean error (0.162). Replacing anisotropic kernels with isotropic ones causes the largest WSS degradation (0.210→0.338), consistent with the elongated geometry of coronary arteries where directional bandwidth flexibility is beneficial. Softplus positivity enforcement produces a comparable WSS increase, suggesting that the sharper gradient of the squared mapping ($g'(x) = 2x$) is

advantageous for learning the precision matrix entries. Sharing kernel parameters between pressure and WSS raises the mean error from 0.162 to 0.187, confirming that the two fields have distinct spatial correlation structures. FiLM and CLS conditioning yield similar pressure errors, but FiLM produces a lower WSS error (0.210 vs. 0.224), likely because the uniform affine modulation couples more directly with the subsequent linear decoder head.

We also test whether simply providing centerline information to an existing baseline is sufficient by augmenting GNOT with a centerline cross-attention branch. On single-vessel data, the added branch degrades performance (0.191 vs. 0.149), while on multi-vessel data it improves over the standard GNOT (0.596 vs. 0.650) yet still far exceeds the error of our RBF decoder. Full results are in Appendix A.5.

## 6 Conclusion

We present a Transformer-Anisotropic RBF framework for predicting wall pressure and shear stress in coronary arteries from centerline geometry, hemodynamic descriptors, and inlet flow rate. The model provides high-fidelity, continuous predictions at low cost, obtaining lower $\ell_2$ errors than all baselines while requiring up to $13.8\times$ fewer FLOPs (at 128 RBF centers vs. GNOT, both evaluated with 2,048 query points). We also introduce a large-scale synthetic coronary hemodynamics dataset (4,200 single-vessel and 4,800 multi-vessel geometries with CFD-derived pressure and WSS), supporting robust training, reproducible benchmarking, and future work on multi-vessel anatomies.

**Limitations and future work.** The training data is entirely synthetic, generated from parametric vessel geometries with idealized boundary conditions (steady-state, Newtonian, rigid walls). While this controlled setup enables reproducible evaluation, it does not capture the full variability of patient-specific anatomies, pulsatile flow, or compliant vessel walls. Importantly, prior work has shown that steady-state CFD closely approximates pulsatile results for pressure-based indices Lo et al. (2019); Nannini et al. (2024); Suk et al. (2024b); nonetheless, extending the framework to time-varying predictions remains an important direction. Future work should explore augmentation strategies, incorporation of patient-derived imaging data (e.g., from coronary CT angiography), and validation against invasive FFR measurements for clinical translation. Currently, our model does not provide uncertainty estimates; incorporating epistemic and aleatoric uncertainty would strengthen reliability for cases outside the training distribution. Cross-dataset evaluation against independently generated coronary hemodynamics data Suk et al. (2021) is left to future work due to representation gaps (missing centerline annotations, unrecorded flow rates, differing conventions); see Appendix A.15 for details. Together, the framework and dataset provide a foundation for fast, anatomically aware hemodynamic surrogate modeling on synthetic coronary geometries, with potential for future extension to patient-specific clinical workflows.

**Broader impact.** This work is a research contribution to computational hemodynamics and is not intended for clinical use. All training and evaluation are performed on synthetic data; predictions have not been validated against clinical measurements. Any future clinical application would require rigorous regulatory approval, prospective validation, uncertainty quantification, and out-of-distribution detection to ensure safe deployment.

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

# A Experimental Details

This appendix provides additional details supporting Section 5, covering CFD simulation setup, data preprocessing, training configuration, model architectures, a centerline input ablation for GNOT, evaluation metrics, FFR evaluation, wall-clock inference timing, patient-disjoint split details, failure-case analysis, and supplementary results.

## A.1 CFD Simulation Setup

All flow simulations use OpenFOAM Jasak (2009) (version 11) with the steady-state incompressible `simpleFoam` solver (SIMPLEC pressure-velocity coupling, 20 non-orthogonal correctors). Blood is modeled as a Newtonian fluid with density $\rho = 1{,}060\,\text{kg/m}^3$ and dynamic viscosity $\mu = 4.0\,\text{mPa·s}$ (kinematic viscosity $\nu = 3.77\,\text{mm}^2/\text{s}$), with rigid no-slip vessel walls and laminar flow, justified by branch Reynolds numbers on the order of $10^2$.

**Mesh generation.** Volume meshes are structured hexahedral grids generated by a custom pipeline that constructs butterfly-topology blocks conforming to the vessel lumen. Meshes contain 100,000 to 300,000 cells depending on vessel complexity (single-vessel: $\sim$120 to 165K; multi-vessel: $\sim$200 to 300K), with 4 boundary-layer cells on vessel walls (expansion ratio 1.0) and 5 layers at inlet/outlet patches. Mesh quality is enforced via non-orthogonality $\leq 65°$, skewness $\leq 4.0$, and cell determinant $\geq 0.02$.

**Boundary conditions.** Outlets prescribe physiological flow splits using `flowRateOutletVelocity`; the inlet velocity is pressure-driven with a reference pressure; walls enforce no-slip velocity and zero-gradient pressure. Inlet flow rates range from 2,500 to 8,000 $\text{mm}^3/\text{s}$ across the dataset.

**Convergence and post-processing.** Simulations are considered converged when pressure residuals drop below $10^{-6}$ (typically reaching $\sim$6$\times$10$^{-8}$) and the global continuity error reaches $\sim$7$\times$10$^{-6}$. Pressure and wall shear stress are computed in kinematic form ($p/\rho$, $\tau_w/\rho$) and converted to physical units (Pa) by multiplying by the fluid density. WSS magnitude is extracted as $|\boldsymbol{\tau}_w|$ at the wall surface mesh vertices via the `wallShearStress` function object.

## A.2 Data Preprocessing

All inputs undergo a fixed preprocessing pipeline before being fed to the model. For each case, surface and centerline positions are centered by subtracting the mean surface position, so that all geometries share a common origin. Surface and centerline coordinates are min-max normalized to $[0, 1]$ using global minimum and maximum values (with a buffer of 1.0) computed across all data splits. Four scalar fields are standardized to zero mean and unit variance using statistics computed on the training split only: surface pressure, surface WSS magnitude, centerline radius, and centerline flowrate. Surface normals and centerline tangent vectors are not normalized. During training, 2,048 surface points are randomly sampled per case. For the RBF model, $M$ centerline points are randomly sampled and sorted by index to preserve spatial ordering along the vessel.

## A.3 Training Configuration

We optimize all models with AdamW (learning rate $10^{-3}$, weight decay $5 \times 10^{-4}$) with batch size 128. We train all models for 5,000 epochs on single-vessel data and 3,000 epochs on multi-vessel data. Single-vessel experiments were run on an NVIDIA Quadro RTX 8000 GPU; multi-vessel experiments were run on an NVIDIA A100 80GB PCIe GPU. On the multi-vessel dataset (3,000 epochs), training takes approximately 7 h for our RBF model (512 centers), 13 h (1,024 centers), 23 h for GNOT, 27 h for ONO, 10 h for Transolver, and 8 h for CenterlinePointNet++ on the A100. The learning rate schedule uses a 500-epoch linear warmup from 0 to full LR, followed by cosine annealing to 0. For all experiments, results are reported as mean±standard deviation over three random seeds (42, 1234, 7890).

For the single-vessel dataset we train on 3,600 cases, validate on 400, and test on 200; for the multi-vessel dataset we train on 4,000 cases, validate on 400, and test on 400. The training objective minimizes

$$\mathcal{L} = \frac{1}{N} \sum_{i=1}^{N} \|\hat{p}_i - p_i\|^2 + \frac{1}{N} \sum_{i=1}^{N} \|\widehat{\text{wss}}_i - \text{wss}_i\|^2, \tag{8}$$

where $\hat{p}_i$ and $\widehat{\text{wss}}_i$ are predictions and $p_i$, $\text{wss}_i$ are targets, all in z-score normalized space. We select the checkpoint with the lowest validation loss.

## A.4    Model Configurations

All baseline models operate directly on surface points and receive only the scalar flow rate as a global condition; our RBF model instead processes centerline points through the transformer encoder and evaluates the learned kernels at the surface query locations. All baselines were reimplemented from the respective official codebases and adapted to our hemodynamic prediction task (surface-based regression of pressure and WSS with FiLM Pérez et al. (2018) flow-rate conditioning). Hyperparameters (depth, width, attention heads) were tuned so that all models have comparable parameter counts ($\sim$2.9M).

**Ours (RBF).**    The encoder receives $M$ centerline points, each described by position and radius $(x, y, z, r)$; the scalar flow rate is injected via FiLM after the encoder. The decoder evaluates the learned anisotropic RBF kernels at 2,048 surface query positions $(x, y, z)$. Architecture: transformer encoder with 5 layers, model dimension 256, feedforward dimension 512, 1 attention head, and dropout 0.1. Centerline tokens are built via Fourier embeddings (hidden dimension 48) projected to the model dimension with a linear layer; learned positional encodings are added over the token sequence. Total: 2.82-3.05M parameters depending on the number of RBF centers (128-1,024).

**ONO Xiao et al. (2023).**    Receives 2,048 surface points $(x, y, z)$ and their normals $(n_x, n_y, n_z)$ as input; the scalar flow rate is injected via FiLM. 4 layers, model dimension 240, 1 attention head, dropout 0.1, with orthogonal projection dimension $\psi_{\text{dim}}$=8 and linear attention. Total: 2.91M parameters.

**GNOT Hao et al. (2023).**    Receives 2,048 surface points $(x, y, z)$ and their normals $(n_x, n_y, n_z)$ as input; the scalar flow rate is broadcast to every point and embedded jointly. 4 layers, model dimension 126, 1 attention head, dropout 0.1, with 4 mixture-of-experts and heterogeneous cross-attention with normalized linear attention. Total: 2.97M parameters.

**Transolver Wu et al. (2024).**    Receives 2,048 surface points $(x, y, z)$ and their normals $(n_x, n_y, n_z)$ as input; the scalar flow rate is injected via FiLM. 4 layers, model dimension 256, 1 attention head, dropout 0.1, with 32 physics-aware slices. Total: 2.93M parameters.

**CenterlinePointNet++[†] Rygiel et al. (2023).**    Adapted from the original architecture designed for scalar vFFR estimation on synthetic coronary arteries. The original model uses 7 CSA/FP blocks with geodesic centerline grouping (shortest-path distance on the centerline graph) and trains separate models for each boundary condition setting. We make the following adaptations for our task: (i) we reduce to 4 CSA/FP blocks to match the $\sim$2.9M parameter budget; (ii) we replace geodesic grouping with $k$-nearest-neighbor grouping in Euclidean space, which approximates centerline-guided neighborhoods while enabling efficient batched GPU computation; (iii) we add FiLM conditioning at the bottleneck to handle variable inlet flow rates within a single model (as the original authors suggested as future work); and (iv) we augment input features with surface normals and relative position to the nearest centerline node (the original uses Euclidean distance to centerline and geodesic distance to inlet). The model operates on 2,048 surface points with 128 centerline nodes and predicts per-point pressure and WSS. Total: 2.91M parameters.

[†]*Adapted from the original; see above for modifications.*

## A.5 Centerline Input Ablation

To test whether simply providing centerline information to an existing baseline is sufficient, we ablate the input configuration of GNOT, which natively supports heterogeneous inputs on different grids via its cross-attention branches Hao et al. (2023). We evaluate three configurations: *normals-only* (surface positions, normals, and flow rate), *centerline-only* (surface positions, centerline positions and radii, and flow rate, no normals), and *both* (all inputs). ONO and Transolver operate exclusively on surface tokens through self-attention and have no mechanism to ingest a separate input sequence, so this ablation is limited to GNOT.

Table 5 reports the results. On single-vessel data, normals-only GNOT achieves the best test error (0.149), while adding the centerline branch degrades performance; the centerline-only variant reaches 0.190 and the combined variant 0.191. This suggests that, for simple tubular geometries, surface normals already encode sufficient geometric information, and that the additional centerline branch introduces optimization difficulty. On multi-vessel data, the trend reverses: the combined variant improves over normals-only GNOT (0.596 vs. 0.650), indicating that explicit centerline geometry becomes beneficial when vessel topology is complex.

Table 5: GNOT input ablation. Test relative $\ell_2$ error for three input configurations. Single-vessel results use 5,000 epochs; multi-vessel results use 3,000 epochs on the standard split. All runs use a single seed to illustrate the architectural design choice; qualitative conclusions are consistent across training regimes.

| Dataset | Input config | Params | Pressure | WSS | Mean |
|---|---|---|---|---|---|
| Single-vessel | Normals only | 2.97M | 0.133 | **0.164** | **0.149** |
| | Centerline only | 2.97M | 0.144 | 0.235 | 0.190 |
| | Both | 3.15M | **0.143** | 0.239 | 0.191 |
| Multi-vessel | Normals only | 2.97M | 0.589 | 0.711 | 0.650 |
| | Centerline only | 2.97M | 0.556 | **0.642** | 0.599 |
| | Both | 3.15M | **0.540** | 0.651 | **0.596** |

## A.6 Evaluation Metrics

**Fractional Flow Reserve (FFR).** FFR is a clinically important hemodynamic index used to assess the functional significance of coronary artery stenoses Pijls et al. (1996). Clinically, it is defined as the ratio of mean distal coronary pressure to mean aortic pressure during maximal hyperemia. In our steady-state CFD simulations, we compute a pointwise pressure ratio at every surface point $\mathbf{x}$:

$$\mathrm{FFR}(\mathbf{x}) = \frac{P(\mathbf{x})}{P_a}, \tag{9}$$

where $P(\mathbf{x}) = \tilde{p}(\mathbf{x})\,\rho/1000 + P_a$ converts the OpenFOAM kinematic pressure output $\tilde{p}$ (in $\mathrm{mm}^2/\mathrm{s}^2$) to absolute pressure using blood density $\rho = 1.06\,\mathrm{g/mL}$, and $P_a$ is the prescribed inlet (aortic) pressure. We report the minimum FFR over the vessel surface, $\mathrm{FFR}_{\min} = \min_{\mathbf{x}} \mathrm{FFR}(\mathbf{x})$, as the per-case summary metric. An FFR value below 0.80 is typically considered indicative of flow-limiting stenosis and is used to guide revascularization decisions Tonino et al. (2009).

**Low-fidelity baseline.** We compute a 1D pressure profile along the vessel centerline using the Hagen-Poiseuille law. For each centerline segment with local radius $r(s)$, the axial pressure gradient is

$$\frac{dP}{ds} = \frac{8\,\mu\,Q}{\pi\,r(s)^4}, \tag{10}$$

where $\mu = 0.004\,\mathrm{Pa \cdot s}$ is the dynamic viscosity and $Q$ is the volumetric flow rate. The cumulative pressure drop is integrated along the centerline to obtain $P(s)$, from which FFR is computed as $(P_a - \Delta P(s))/P_a$. For multi-vessel cases, flow is split equally among downstream branches at each bifurcation.

## A.7 FFR Evaluation

Tables 6 and 7 report FFR classification metrics on the test set for single-vessel and multi-vessel data, respectively.

**Single-vessel**   (Table 6; threshold FFR < 0.8; 134 positive, 66 negative). Our model (512 RBFs) achieves the lowest MAE (0.014±0.001) with sensitivity 97.0±0.6% and specificity 91.9±0.7%. GNOT obtains comparable classification performance (sensitivity 98.0±0.7%, specificity 86.4±4.5%) but with nearly double the MAE (0.027±0.001). The low-fidelity 1D Poiseuille baseline predicts nearly all cases as non-significant (FFR≥0.8), yielding only 1.5% sensitivity despite 100% specificity.

**Multi-vessel**   (Table 7; threshold FFR < 0.5; 273 positive, 127 negative). All 400 multi-vessel test cases have ground-truth FFR < 0.8, reflecting the high prevalence of hemodynamically significant stenoses in the ImageCAS-derived dataset. Classification at the standard 0.8 threshold is therefore uninformative (all models achieve ≥97.5% accuracy trivially). We instead evaluate at FFR < 0.5, which distinguishes severe from moderate stenoses and yields a more balanced class distribution. At this threshold, our model (1024 RBFs) achieves 87.7±0.3% accuracy with 86.1±1.1% sensitivity and 91.3±3.1% specificity, while all baselines perform near chance level (53 to 57% accuracy). The MAE of our model (0.082±0.002) is more than 2.5× lower than the best baseline, confirming that our predictions are quantitatively closer to the ground truth across the full FFR range.

Table 6: FFR evaluation on single-vessel test data (threshold FFR < 0.8; 134 positive, 66 negative cases). Results are mean±std over 3 seeds.

| Method | MAE (↓) | Sensitivity | Specificity | Accuracy | F1 |
|---|---|---|---|---|---|
| Low-Fidelity | 0.187 | 0.015 | **1.000** | 0.340 | 0.029 |
| GNOT | 0.027±0.001 | 0.980±0.007 | 0.864±0.045 | 0.942±0.014 | 0.958±0.010 |
| Transolver | 0.044±0.013 | 0.963±0.024 | 0.672±0.186 | 0.867±0.062 | 0.908±0.039 |
| ONO | 0.031±0.011 | **0.993±0.000** | 0.838±0.052 | 0.942±0.017 | 0.958±0.012 |
| Ours | **0.014±0.001** | 0.970±0.006 | 0.919±0.007 | **0.953±0.002** | **0.965±0.002** |

Table 7: FFR evaluation on multi-vessel test data (threshold FFR < 0.5; 273 positive, 127 negative cases). All test cases have FFR < 0.8; we use 0.5 to distinguish severe from moderate stenoses. Results are mean±std over 3 seeds (42, 1234, 7890).

| Method | MAE (↓) | Sensitivity | Specificity | Accuracy | F1 |
|---|---|---|---|---|---|
| Low-Fidelity | 0.523 | 0.004 | **1.000** | 0.320 | 0.007 |
| GNOT | 0.231±0.002 | 0.390±0.002 | 0.850±0.008 | 0.536±0.004 | 0.535±0.003 |
| Transolver | 0.223±0.006 | 0.462±0.000 | 0.787±0.024 | 0.565±0.008 | 0.592±0.004 |
| ONO | 0.212±0.006 | 0.451±0.084 | 0.756±0.047 | 0.547±0.042 | 0.571±0.069 |
| Ours | **0.082±0.002** | **0.861±0.011** | 0.913±0.031 | **0.877±0.003** | **0.906±0.001** |

Figure 9 shows a Bland-Altman analysis of FFR agreement for our best model on each dataset. On single-vessel data, the mean bias is near zero with narrow limits of agreement, confirming accurate FFR estimation. On multi-vessel data, the bias remains small but the limits of agreement are wider, reflecting the greater difficulty of the multi-vessel prediction task.

## A.8 Wall-Clock Inference Timing

Table 8 reports wall-clock inference latency on an NVIDIA A100 80GB GPU with 2,048 surface query points and batch size 32. Each model was run for 10 warm-up forward passes followed by 100 timed passes with explicit `cuda.synchronize()`; we report per-sample time. A batch size of 32 is used to ensure the GPU is

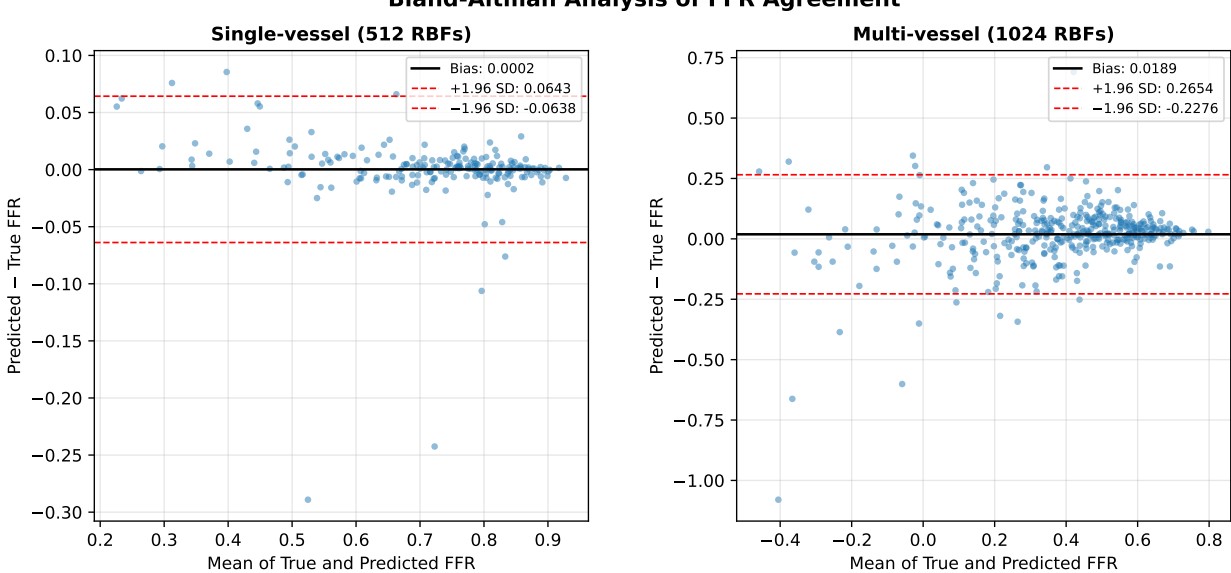

Figure 9: Bland-Altman plots of predicted vs. true FFR for our model on single-vessel (left, 512 RBFs) and multi-vessel (right, 1024 RBFs) test sets. Solid line: mean bias; dashed lines: ±1.96 standard deviation limits of agreement.

compute-saturated, so that wall-clock differences reflect actual computational cost rather than kernel-launch overhead. At 128 RBFs, our model runs in 0.18 ms/sample, 14× faster than GNOT (2.54 ms), 12.8× faster than ONO (2.31 ms), and 4.5× faster than Transolver (0.81 ms), consistent with the theoretical FLOP ratios. Even at 1,024 RBFs (1.32 ms), our model remains faster than all baselines.

Table 8: Wall-clock inference time on an NVIDIA A100 GPU with 2,048 surface points and batch size 32.

| Method | Params (M) | GFLOPs | Time (ms/sample) |
|---|---|---|---|
| GNOT Hao et al. (2023) | 2.97 | 7.31 | 2.54 |
| Transolver Wu et al. (2024) | 2.93 | 4.59 | 0.81 |
| ONO Xiao et al. (2023) | 2.91 | 10.37 | 2.31 |
| CenterlinePointNet++[†] Rygiel et al. (2023) | 2.91 | 3.32 | 3.87 |
| Ours (128 RBFs) | 2.82 | 0.53 | **0.18** |
| Ours (256 RBFs) | 2.85 | 1.22 | 0.33 |
| Ours (512 RBFs) | 2.92 | 3.12 | 0.63 |
| Ours (1024 RBFs) | 3.05 | 8.93 | 1.32 |

## A.9 Encoder vs. Decoder Cost Decomposition

Table 9 decomposes the total FLOPs of our model into the encoder (transformer processing of $M$ centerline tokens, including FiLM conditioning and output projection) and the decoder (RBF kernel evaluation at $N$=2,048 surface query points). The encoder dominates at all RBF counts (96 to 98%), while the decoder scales linearly with both $M$ and $N$. Doubling the number of query points from 2,048 to 4,096 adds only 0.020 GFLOPs at $M$=128 (a 3.8% increase), explaining the near-invariance of total cost to query resolution.

Table 9: Encoder vs. decoder GFLOPs for our model ($N$=2,048 query points).

| $M$ (RBFs) | Encoder (G) | Decoder (G) | Total (G) | Encoder % |
| --- | --- | --- | --- | --- |
| 64 | 0.233 | 0.010 | 0.243 | 95.8 |
| 128 | 0.508 | 0.020 | 0.528 | 96.2 |
| 256 | 1.183 | 0.040 | 1.224 | 96.7 |
| 512 | 3.038 | 0.081 | 3.119 | 97.4 |
| 1024 | 8.764 | 0.162 | 8.925 | 98.2 |

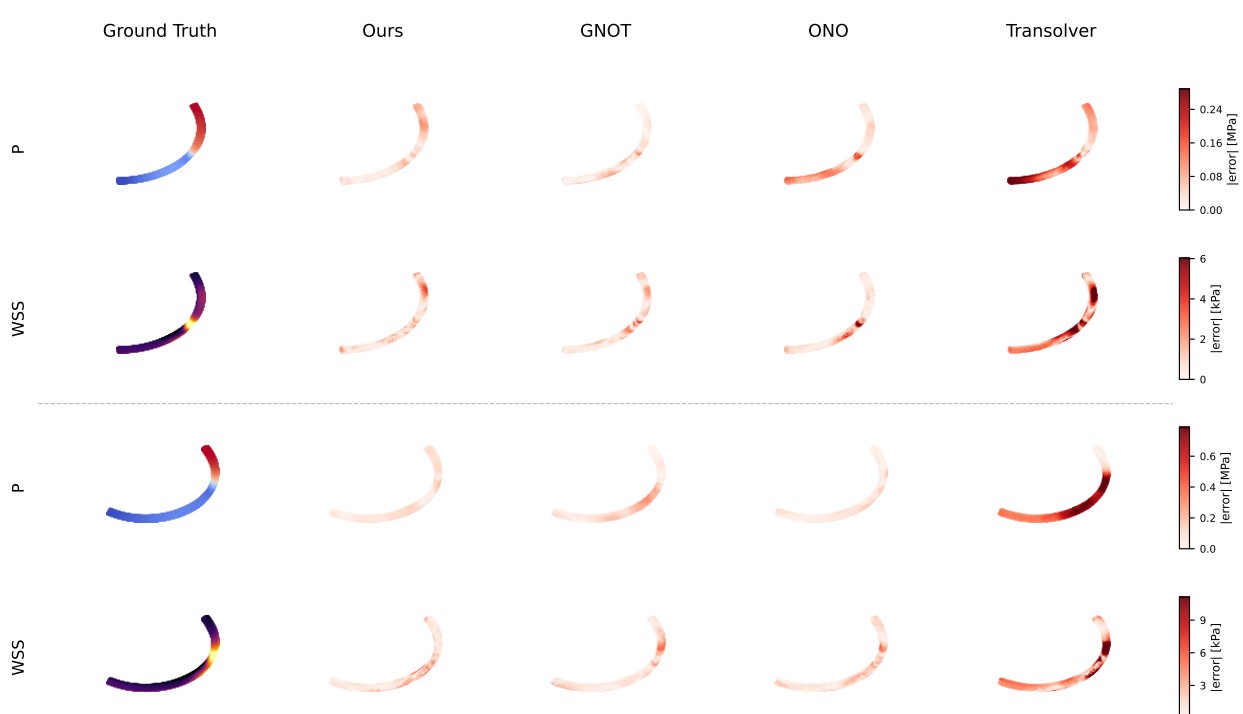

Figure 10: Pointwise absolute prediction error on two single-vessel test geometries. The first column shows the ground-truth field; columns 2 to 5 display $|\hat{y} - y|$ for each model on a shared color scale. Our model (128 RBFs) attains relative $\ell_2$ errors of 3.51%/5.83% and 2.74%/5.18% (P/WSS). GNOT (3.15%/5.95%, 3.87%/5.83%) and ONO (5.29%/8.16%, 2.32%/5.11%) are competitive on single-vessel data, while Transolver shows higher error (11.54%/20.85%, 17.19%/21.65%).

### A.10   Qualitative Single-Vessel Comparison

Figure 10 shows the pointwise absolute prediction error on two single-vessel test cases. On this simpler geometry, our model, GNOT, and ONO all achieve low errors, while Transolver lags behind on both examples.

### A.11   Circumferential WSS Reconstruction

A natural concern with centerline-anchored RBF kernels is whether they can capture circumferential variation in WSS, which is non-axisymmetric near stenoses. Figure 11 plots the predicted and ground-truth WSS as a function of circumferential angle $\theta$ at two axial stations of a single-vessel test case: (left) the stenotic throat, where WSS is elevated and varies around the circumference, and (right) a healthy upstream section. At both stations, the anisotropic RBF decoder closely tracks the ground-truth circumferential profile, confirming that the learned precision matrices adapt to capture non-axisymmetric WSS patterns despite the kernels being anchored on the 1D centerline.

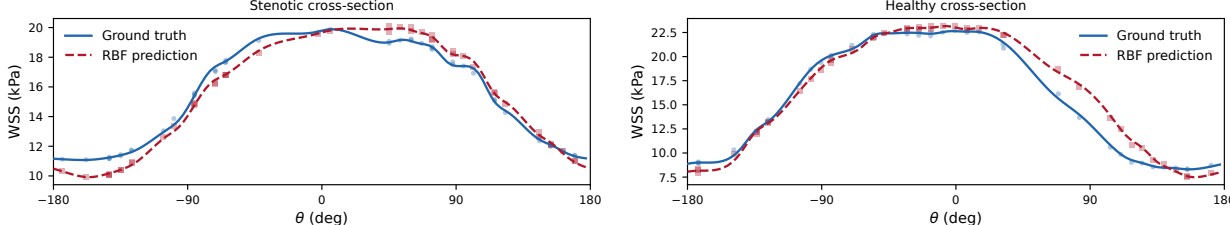

Figure 11: Circumferential WSS profile at two axial stations of a single-vessel test case. Left: stenotic throat (minimum radius); right: healthy upstream section. Scatter points show individual surface samples; curves are smoothed fits. The anisotropic RBF decoder captures the circumferential WSS variation at both stations.

## A.12 Training and Validation Results

Tables 10 and 11 report training-set errors at the final training epoch for single-vessel and multi-vessel datasets, respectively. Comparing with the test results in Table 1 and Table 2 reveals the generalization gap for each model. On single-vessel data, GNOT achieves the lowest training error ($0.031\pm0.022$) but degrades to $0.149\pm0.010$ at test time ($\sim5\times$ gap), indicating overfitting. ONO shows a similar pattern ($0.041\pm0.016 \rightarrow 0.145\pm0.039$, $\sim4\times$ gap). Transolver shows high variance and does not converge well with some seeds, yielding training error of $0.125\pm0.108$. Our model (512 RBFs) reaches a training error of $0.034\pm0.004$ with the smallest test-time degradation ($0.116\pm0.012$, $\sim3.4\times$ gap), demonstrating stronger generalization.

On multi-vessel data, the baselines converge to moderate training errors by the final epoch (GNOT 0.215, Transolver 0.207, ONO 0.246) but degrade substantially at test time (all $> 0.64$), with best-validation checkpoints selected at early epochs. Our model scales consistently with the number of RBF bases, reaching a training error of 0.138 at 1,024 RBFs with a test error of 0.304 ($\sim2.2\times$ gap), substantially outperforming all baselines.

Table 10: Single-vessel training set (3,600 cases). Relative $\ell_2$ error (mean$\pm$std over 3 seeds) at the final training epoch.

| Method | Pressure | WSS | Mean |
|---|---|---|---|
| GNOT Hao et al. (2023) | $0.022\pm0.017$ | $\mathbf{0.040\pm0.028}$ | $\mathbf{0.031\pm0.022}$ |
| Transolver Wu et al. (2024) | $0.100\pm0.092$ | $0.150\pm0.124$ | $0.125\pm0.108$ |
| ONO Xiao et al. (2023) | $0.029\pm0.013$ | $0.053\pm0.019$ | $0.041\pm0.016$ |
| Ours (64 RBFs) | $0.038\pm0.001$ | $0.095\pm0.009$ | $0.067\pm0.004$ |
| Ours (128 RBFs) | $0.031\pm0.004$ | $0.068\pm0.001$ | $0.050\pm0.003$ |
| Ours (256 RBFs) | $0.022\pm0.001$ | $0.053\pm0.003$ | $0.037\pm0.002$ |
| Ours (512 RBFs) | $\mathbf{0.021\pm0.002}$ | $0.048\pm0.006$ | $0.034\pm0.004$ |

Tables 12 and 13 report validation-set errors corresponding to the test results in Table 1 and Table 2, respectively.

## A.13 Patient-Disjoint Split Details

The multi-vessel dataset is derived from 140 unique patient anatomies, each with multiple physiological variants (different stenosis placements and boundary conditions). The standard data split (used for Table 2) assigns cases randomly, allowing variants from the same patient anatomy to appear in different splits. While this ensures diverse representation, it creates potential data leakage since variants share the same underlying vessel geometry.

Table 11: Multi-vessel training set (4,000 cases). Relative $\ell_2$ error (mean±std over 3 seeds) at the final training epoch.

| Method | Pressure | WSS | Mean |
|---|---|---|---|
| GNOT Hao et al. (2023) | 0.158±0.005 | 0.273±0.004 | 0.215±0.004 |
| Transolver Wu et al. (2024) | 0.147±0.002 | 0.267±0.002 | 0.207±0.001 |
| ONO Xiao et al. (2023) | 0.175±0.007 | 0.316±0.005 | 0.246±0.005 |
| CenterlinePointNet++[†] Rygiel et al. (2023) | 0.072±0.000 | **0.126±0.000** | **0.099±0.000** |
| Ours (128 RBFs) | 0.145±0.003 | 0.392±0.002 | 0.269±0.002 |
| Ours (256 RBFs) | 0.106±0.003 | 0.322±0.003 | 0.214±0.003 |
| Ours (512 RBFs) | 0.081±0.002 | 0.264±0.004 | 0.172±0.003 |
| Ours (1024 RBFs) | **0.066±0.001** | 0.211±0.002 | 0.138±0.001 |

Table 12: Single-vessel validation set (400 cases). Relative $\ell_2$ error (mean±std over 3 seeds).

| Method | Pressure | WSS | Mean |
|---|---|---|---|
| Low-Fidelity | 0.639 | 0.488 | 0.563 |
| GNOT Hao et al. (2023) | 0.116±0.010 | 0.156±0.009 | 0.136±0.009 |
| Transolver Wu et al. (2024) | 0.170±0.045 | 0.240±0.073 | 0.205±0.059 |
| ONO Xiao et al. (2023) | 0.124±0.029 | 0.133±0.025 | 0.128±0.027 |
| Ours (64 RBFs) | 0.109±0.006 | 0.153±0.003 | 0.131±0.003 |
| Ours (128 RBFs) | 0.104±0.001 | 0.140±0.002 | 0.122±0.001 |
| Ours (256 RBFs) | 0.096±0.002 | 0.136±0.008 | 0.116±0.005 |
| Ours (512 RBFs) | **0.091±0.004** | **0.126±0.005** | **0.108±0.004** |

To address this, we constructed a patient-disjoint re-split. All variants originating from a single patient anatomy are assigned exclusively to one of {train, val, test}, guaranteeing zero overlap of patient anatomies between splits. The split was performed by grouping cases by patient ID and allocating groups to maintain proportional representation of vessel types (LCA/RCA).

The patient-disjoint split sizes are:

- **Training**: 3,989 cases (116 unique patient anatomies)

- **Validation**: 415 cases (12 unique patient anatomies)

- **Test**: 396 cases (12 unique patient anatomies)

Normalization statistics for this split were recomputed based solely on its training set.

Tables 14 and 15 report validation-set and training-set errors for the patient-disjoint split, complementing the test results in Table 3.

### A.14   Failure Case Analysis

To understand the limits of the proposed model, we characterize the error distribution on both multi-vessel test sets and visualize the highest-error cases on the standard split, stratified by vessel type.

**Error distribution.**   On the standard split, the per-case mean relative $\ell_2$ error has a benign tail: the 95th percentile is 0.41 and only 2% of cases exceed 0.5. On the patient-disjoint split, the tail is substantially heavier (95th percentile 1.35; 8% of cases exceed 1.0), reflecting the greater difficulty of generalizing to entirely unseen patient anatomies.

Table 13: Multi-vessel validation set (400 cases). Relative $\ell_2$ error (mean±std over 3 seeds).

| Method | Pressure | WSS | Mean |
|---|---|---|---|
| Low-Fidelity | 0.852 | 0.659 | 0.756 |
| GNOT Hao et al. (2023) | 0.551±0.008 | 0.735±0.004 | 0.643±0.006 |
| Transolver Wu et al. (2024) | 0.563±0.014 | 0.727±0.002 | 0.645±0.006 |
| ONO Xiao et al. (2023) | 0.571±0.006 | 0.744±0.006 | 0.658±0.005 |
| CenterlinePointNet++[†] Rygiel et al. (2023) | 0.410±0.007 | **0.324±0.004** | 0.367±0.001 |
| Ours (128 RBFs) | 0.265±0.011 | 0.455±0.010 | 0.360±0.010 |
| Ours (256 RBFs) | 0.215±0.007 | 0.406±0.007 | 0.310±0.007 |
| Ours (512 RBFs) | 0.196±0.010 | 0.384±0.007 | 0.290±0.008 |
| Ours (1024 RBFs) | **0.188±0.004** | 0.370±0.003 | **0.279±0.003** |

Table 14: Patient-disjoint validation set (415 cases). Relative $\ell_2$ error (mean±std over 3 seeds).

| Method | Pressure | WSS | Mean |
|---|---|---|---|
| GNOT Hao et al. (2023) | 0.601±0.013 | 0.714±0.009 | 0.657±0.011 |
| Transolver Wu et al. (2024) | 0.592±0.018 | 0.701±0.012 | 0.647±0.015 |
| ONO Xiao et al. (2023) | 0.585±0.010 | 0.702±0.006 | 0.644±0.008 |
| Ours (128 RBFs) | **0.185±0.012** | **0.428±0.009** | **0.306±0.010** |

**Geometry correlates.** We compute Pearson correlations between per-case error and geometric features extracted from the centerline representation. On the standard split, the number of branches ($r$=0.26), maximum tree depth ($r$=0.26), and total centerline length ($r$=0.26) are the strongest positive correlates, while minimum radius shows a weak negative correlation ($r$=−0.18). On the patient-disjoint split, minimum vessel radius becomes the dominant factor ($r$=−0.35), indicating that tight stenoses in unseen anatomies are the hardest to predict. Left coronary arteries (LCA) exhibit consistently higher errors than right coronary arteries (RCA) on both splits (standard: 0.31 vs. 0.24; patient-disjoint: 0.75 vs. 0.51), attributable to LCA's greater branching complexity (mean 6.8 branches vs. 3.0).

**Cross-model consistency.** Failures are highly consistent across random seeds for a given architecture (cross-seed $r$=0.88 on the standard split), confirming that they are geometry-dependent rather than stochastic. However, per-case errors are only weakly correlated across architectures (our model vs. GNOT/Transolver: $r$≈0.25 to 0.30), indicating that different model families fail on partially different subsets of cases.

**Visualization.** Figure 12 shows the ground-truth pressure and WSS fields alongside pointwise absolute prediction errors for our model (1024 RBFs) on the three highest-error LCA and RCA test cases from the standard split. Among the LCA failures, case 596 ($\ell_2$=1.03) is a complex 9-branch geometry where pressure error concentrates at the distal branches, while cases 805 ($\ell_2$=0.76) and 381 ($\ell_2$=0.70) show errors localized near tight stenoses and bifurcations. The RCA failures exhibit lower overall error (mean $\ell_2$ between 0.41 and 0.62), with errors again concentrated at stenosis sites and branch points. In all cases, WSS errors are spatially correlated with regions of high WSS gradient.

## A.15 Cross-Dataset Evaluation

We explored cross-dataset evaluation using the publicly available coronary hemodynamics dataset of Suk et al. Suk et al. (2021), which provides ∼4,000 coronary vessel geometries with OpenFOAM-computed steady-state pressure and WSS. However, several representation gaps prevent a direct zero-shot evaluation: (i) the dataset stores triangulated surface meshes (∼10,000 to 25,000 nodes) without centerline annotations, which are a required input to our model; extracting centerlines, radii, tangent vectors, and branch IDs from closed surface meshes is feasible but introduces a non-trivial preprocessing pipeline; (ii) inlet flow rate,

Table 15: Patient-disjoint training set (3,989 cases). Relative $\ell_2$ error (mean±std over 3 seeds) at the final training epoch.

| Method | Pressure | WSS | Mean |
|---|---|---|---|
| GNOT Hao et al. (2023) | 0.158±0.005 | 0.273±0.004 | 0.215±0.002 |
| Transolver Wu et al. (2024) | 0.147±0.002 | **0.267±0.002** | **0.207±0.002** |
| ONO Xiao et al. (2023) | 0.175±0.007 | 0.316±0.005 | 0.246±0.005 |
| Ours (128 RBFs) | **0.145±0.003** | 0.392±0.002 | 0.269±0.002 |

used by our model via FiLM conditioning, is not recorded and would need to be estimated from pressure gradients; and (iii) unit conventions and meshing strategies differ between the two datasets. Bridging these representation gaps constitutes a standalone engineering effort that we leave to future work.

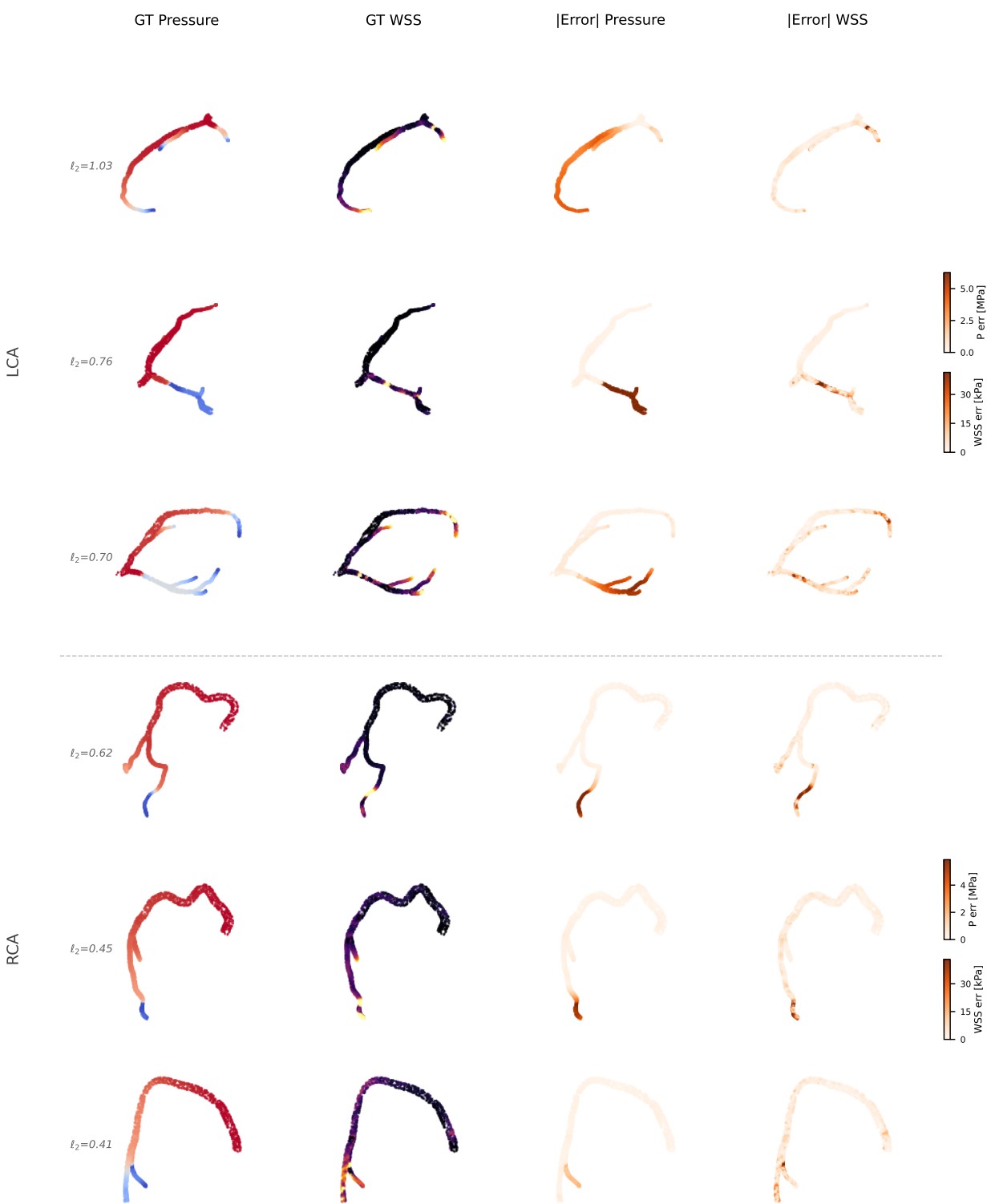

Figure 12: Failure case analysis on the standard multi-vessel test set. Ground-truth pressure and WSS fields (columns 1–2) and pointwise absolute prediction error of our model (1024 RBFs, columns 3–4) for the three highest-error LCA cases (top, patients 596, 805, 381) and three highest-error RCA cases (bottom, patients 372, 371, 818). Error color scales are shared within each vessel-type group. Row labels show the mean relative $\ell_2$ error. Errors concentrate at stenosis sites, branch points, and distal branches of complex LCA geometries.

