# OpenReview forum: "From Centerlines to Hemodynamics: Anisotropic RBF Decoders for Coronary Arteries"
_TMLR — Decision pending for TMLR_

### Review · Reviewer_vg7j · 2026-06-10

**Summary Of Contributions:**

This paper proposes a novel framework (Transformer encoder combined with an anisotropic RBF decoder) to predict coronary hemodynamics, such as pressure and wall shear stress. Instead of treating blood vessels as standard 3D meshes, this work reduces the dimensionality by leveraging 1D vessel centerlines for inference, which significantly reduces computational costs. Additionally, the authors introduce two synthetic datasets (single-vessel and multi-vessel) paired with CFD simulations.

Strengths:
1. The architectural design is clever. Using 1D centerlines combined with an RBF decoder to reconstruct continuous 3D fields effectively addresses the computational bottleneck of 3D mesh-based methods.
2. The empirical comparisons are thorough. The model demonstrates clear advantages in both predictive accuracy and inference efficiency (FLOPs) compared to recent neural operator baselines like GNOT, ONO, and Transolver.

Weaknesses:
1. As I am not an expert in computational fluid dynamics or clinical cardiology, it is difficult for me to accurately gauge how limiting the idealized physical assumptions (steady-state, Newtonian fluid, rigid walls) are for real-world clinical applications.
2. The model is currently only validated on synthetic datasets, lacking proof of generalization to real clinical data distributions.

**Audience:**

Yes

**Audience Explanation:**

Although this is a highly specific medical physics simulation application, the core idea—reducing a 3D spatial problem into a 1D sequence processing task based on tubular topological priors—is highly relevant and inspiring for TMLR readers interested in AI for Science, neural PDE solvers, and geometric deep learning.

**Broader Impact Concerns:**

None. The authors have appropriately addressed this by explicitly stating in the conclusion that the method is validated only on synthetic data, is strictly not intended for clinical use, and highlighting the necessary validation steps required for any future clinical translation.

**Claims And Evidence:**

Yes

**Claims Explanation:**

The experimental evidence is sufficient and clear. The authors provide detailed quantitative comparisons of prediction errors (relative L2 error) and computational costs (GFLOPs, wall-clock inference time) across both single-vessel and multi-vessel datasets. The figures and tables are intuitive, well-supporting the model's claims regarding efficiency and accuracy improvements over the baselines.

**Requested Changes:**

The authors mention in the limitations that the current results are from single training runs. Given the relatively small parameter size of the models (~2.9M), I highly recommend adding experiments with multiple random seeds and reporting error bars or variance to prove the stability and reproducibility of the model's performance.

Besides, an optional change is that I suggest adding a brief explanation in the discussion section for non-domain experts regarding the actual gap between the current idealized synthetic setup (steady-state) and real "pulsatile flow" clinical data. This would help readers without a medical background better understand the applicability boundaries of this method.

---

> ### Author Response · Authors · 2026-07-03
>
> We thank the reviewer for their positive assessment and helpful suggestions. Both have been addressed.
>
> **1. "I highly recommend adding experiments with multiple random seeds and reporting error bars or variance."**
>
> Good suggestion. Main results now report mean and std over 3 seeds (42, 1234, 7890), confirming stability across configurations.
>
> **2. "I suggest adding a brief explanation... regarding the actual gap between the current idealized synthetic setup (steady-state) and real 'pulsatile flow' clinical data."**
>
> We appreciate this suggestion. It helps make the paper more accessible. We added a discussion in Section 6 citing three studies that support the steady-state approximation for pressure-based indices [1,2,3], while acknowledging that extension to pulsatile flow remains an important direction.
>
>
> [1] Lo et al. "Impact of inflow boundary conditions on the calculation of CT-based FFR." Fluids, 2019.
>
> [2] Nannini et al. "An automated and time efficient framework for simulation of coronary blood flow under steady and pulsatile conditions." Computer Methods and Programs in Biomedicine, 2024.
>
> [3] Suk et al. "Deep vectorised operators for pulsatile hemodynamics estimation in coronary arteries from a steady-state prior." Computer Methods and Programs in Biomedicine, 2024.

---

### Review · Reviewer_DVHp · 2026-06-11

**Summary Of Contributions:**

### Summary

The paper proposes a surrogate model for coronary hemodynamics that uses the vessel centerline, local radii, and inlet flow rate as input. A transformer encodes the 1D centerline sequence, and an anisotropic RBF decoder predicts continuous wall pressure and WSS fields at arbitrary surface query points. The authors also introduce synthetic single-vessel and ImageCAS-derived multi-vessel CFD datasets, and compare against GNOT, Transolver, and ONO.

The main idea is compelling: coronary arteries are naturally tubular, so a centerline-based representation is a sensible inductive bias. The reported multi-vessel results are strong, and the efficiency numbers are attractive. I am positive about the direction, but several details need clarification before the strongest claims are fully convincing.

### Strengths

- The architecture is well motivated by the geometry of the problem. Anchoring anisotropic kernels on the centerline is more natural than treating the surface as an unstructured point cloud.
- The empirical results are encouraging, especially on the multi-vessel dataset where the proposed method substantially outperforms the neural-operator baselines.
- The paper evaluates both accuracy and efficiency, and the ablation study supports the importance of anisotropic kernels and separate pressure/WSS kernel parameters.
- The limitations section is reasonably honest about synthetic data, steady-state assumptions, and lack of clinical validation.

### Weaknesses

My main concern is the dataset split. The multi-vessel dataset is generated from 141 patient anatomies, with multiple stenosis and boundary-condition variants per base anatomy. The paper says the final dataset has fixed train/validation/test splits, but it does not clearly state whether splits are patient/anatomy-disjoint. If variants of the same base anatomy appear in both train and test, the reported multi-vessel generalization results would be much less meaningful.

The CFD setup also needs more detail. The paper uses steady-state, Newtonian, rigid-wall OpenFOAM simulations, which is acceptable for a surrogate benchmark, but the boundary conditions, units, mesh quality, and convergence criteria are under-specified. In particular, the reported outlet velocity units should be checked and justified. Since the paper is in a medical domain, benchmark validity matters as much as model accuracy.

The experimental evidence is otherwise solid but somewhat narrow. All results are from single training runs, so it is hard to tell how stable the improvements are. The baselines are reasonable general neural operators, but the comparison would be stronger with at least one domain-specific hemodynamics or centerline/surface baseline. Finally, the clinical language should be toned down: the work is a synthetic CFD surrogate study, not evidence of clinical readiness.

**Audience:**

Yes

**Audience Explanation:**

As mentioned in the summary of contribution

**Broader Impact Concerns:**

The broader-impact statement is mostly adequate. Because this is a high-stakes medical application, I would add an explicit note that any future use would require clinical validation, uncertainty quantification, out-of-distribution checks, and regulatory review.

**Claims And Evidence:**

Yes

**Claims Explanation:**

As mentioned in the summary of contribution

**Requested Changes:**

- Clarify whether the multi-vessel split is patient/anatomy-disjoint. If it is not, add an anatomy-disjoint evaluation.
- Add enough CFD detail to assess benchmark quality: boundary conditions, units, mesh resolution, convergence checks, and WSS/pressure post-processing.
- Add uncertainty estimates or multi-seed results for the main quantitative tables.
- Adjust the clinical framing in the abstract/introduction. The model is not clinically validated and should not be presented as a near-deployable FFR tool.
- Add a domain-specific baseline or explain clearly why existing coronary surrogate models are not comparable.
- Expand the resolution-scaling analysis and separate encoder cost from decoder/query-point cost.

---

> ### Author Response · Authors · 2026-07-03
>
> We thank the reviewer for their precise and helpful feedback. All requested changes have been addressed in the revision.
>
>
> **1. "Clarify whether the multi-vessel split is patient/anatomy-disjoint. If it is not, add an anatomy-disjoint evaluation."**
>
> Thank you for raising this. The standard split is not anatomy-disjoint. We now provide a patient-disjoint re-split (3,989/415/396) where all variants from one patient go to exactly one split (Appendix A.13). Under this evaluation (Table 3), our model (0.558±0.006) outperforms all baselines (≥0.710).
>
> **2. "Add enough CFD detail to assess benchmark quality: boundary conditions, units, mesh resolution, convergence checks, and WSS/pressure post-processing."**
>
> We agree this was under-specified. Appendix A.1 now covers mesh generation, boundary conditions, convergence criteria, and post-processing in full detail.
>
> **3. "Add uncertainty estimates or multi-seed results for the main quantitative tables."**
>
> Good suggestion. All tables now report mean±std over 3 random seeds (42, 1234, 7890).
>
> **4. "Adjust the clinical framing in the abstract/introduction."**
>
> Agreed. The abstract now says "learned surrogate for fast prediction of CFD-simulated coronary hemodynamics".
>
> **5. "Add a domain-specific baseline or explain clearly why existing coronary surrogate models are not comparable."**
>
> Thank you for this suggestion. We added CenterlinePointNet++ [1], adapted for spatially resolved field prediction with comparable parameters (2.91M). It outperforms generic baselines on multi-vessel data (0.393 vs. >0.64), confirming the benefit of centerline-aware architectures. Details in Appendix A.4 and Table 2.
>
> **6. "Expand the resolution-scaling analysis and separate encoder cost from decoder/query-point cost."**
>
> Good point. Added Appendix A.9 (Table 9): the encoder accounts for 96–98% of FLOPs; the decoder contributes 2 to 4% and is the only component scaling with query count.
>
> **7. Broader impact: "add an explicit note that any future use would require clinical validation, uncertainty quantification, out-of-distribution checks, and regulatory review."**
>
> We agree this was missing. The broader impact statement now includes: "rigorous regulatory approval, prospective validation, uncertainty quantification, and out-of-distribution detection".
>
>
> [1] Rygiel et al. "CenterlinePointNet++: A new point cloud based architecture for coronary artery pressure drop and vFFR estimation". MICCAI 2023.

---

### Review · Reviewer_cg6m · 2026-06-18

**Summary Of Contributions:**

### Summary

The paper studies fast surrogate modeling for coronary hemodynamics. Given a coronary vessel represented by a 1D centerline with radius information, together with an inlet flow rate, the goal is to predict continuous wall-based hemodynamic fields, mainly pressure and WSS magnitude, at arbitrary surface query points. The paper proposes a Transformer-Anisotropic RBF Network. A transformer encoder processes the centerline geometry using Fourier positional embeddings, the inlet flow rate is injected through FiLM conditioning, and an anisotropic radial basis function decoder predicts continuous pressure and WSS magnitude fields by aggregating learned kernels centered along the vessel. The main motivation is that coronary arteries are tubular structures with a natural low-dimensional centerline representation, while many existing neural-operator baselines operate directly on surface or volumetric discretizations. To train and evaluate the model, the authors introduce two CFD-generated datasets. The method is compared against neural-operator baselines including GNOT, Transolver, and ONO. Empirically, the proposed method obtains lower errors for pressure and WSS magnitude on both datasets, with particularly large gains on the more complex multi-vessel setting. The paper also reports improved FFR prediction and better computational efficiency, including substantially fewer FLOPs at lower RBF counts and near-invariant inference cost with respect to the number of query surface points.

Overall, the manuscript presents an interesting method for an important and active scientific machine-learning task. The proposed approach is intuitive and achieves significant gains on the proposed datasets. However, the motivation for why faster hemodynamic prediction is needed should be better explained, and the current validation is not sufficient for the claims made in the paper.

---

### Strengths

1. **Interesting geometry-aware formulation.** The paper exploits the natural 1D centerline structure of coronary arteries rather than treating the vessel only as an unstructured surface or volumetric point cloud. This is a sensible inductive bias for tubular vascular geometries.

2. **Continuous, mesh-independent decoding.** The anisotropic RBF decoder allows pressure and WSS to be evaluated at arbitrary wall locations. This is useful because the output is not tied to a fixed surface discretization and can potentially support different mesh resolutions.

3. **Good accuracy-efficiency tradeoff.** The method appears cheaper than the neural-operator baselines at lower RBF counts, while still outperforming them. The near-invariance of inference cost to the number of query surface points is a meaningful practical advantage.

4. **Useful dataset contribution.** The paper introduces relatively large paired CFD datasets for coronary hemodynamics, including both controlled single-vessel geometries and more realistic multi-vessel anatomies. This could be valuable for future work, especially if the dataset and evaluation protocol are strengthened according to the suggestions below.

---

### Weaknesses

**1. The motivation for the practical need of the method should be clarified.** The paper motivates the work by stating that CFD has limited applicability in time-sensitive clinical workflows because of its computational cost. However, it is not explained concretely what clinical scenarios require this level of speed, how common these scenarios are, or what time constraints are practically relevant. Clarifying this point is important for understanding the significance of the proposed efficiency gains..

**2. The clinical and “non-invasive” framing is stronger than the current validation supports.** The paper presents the method as a framework for fast, non-invasive coronary hemodynamics prediction. However, all training and evaluation targets are obtained from synthetic CFD simulations, and the model is not validated against patient measurements such as invasive FFR, measured pressure, or clinically derived WSS. Simulation-based supervision is understandable for dense hemodynamic fields, but the current experiments establish agreement only with the authors’ simulation pipeline, not clinical accuracy. If suitable patient data are accessible, the paper would be strengthened by at least a limited validation against real measurements. Otherwise, the authors should narrow the claims and describe the method as a surrogate for simulated coronary hemodynamics rather than as a validated non-invasive clinical tool.

**3. The paper provides insufficient evidence of generalization beyond the training distribution and simulation pipeline.** The reported train and test cases appear to be generated using the same geometry-generation procedure, OpenFOAM setup, physical assumptions, and ranges of simulation parameters. Consequently, the experiments primarily evaluate interpolation within a single synthetic distribution rather than generalization to meaningfully different geometries or physical settings. This is a central concern because a useful surrogate model should remain accurate when the anatomy, boundary conditions, simulation assumptions, or input quality differ from those seen during training.

I recommend adding the following generalization experiments:

(a) **Out-of-distribution simulation parameters and physical assumptions.** Construct train and test distributions that differ explicitly. For example, hold out ranges of stenosis severity, stenosis location, inlet flow rate, vessel radius, length, curvature, or branching complexity. The authors should also test changes in boundary conditions and, where computationally feasible, more substantial changes such as pulsatile instead of steady-state flow, alternative outlet conditions, non-Newtonian fluid models, or compliant instead of rigid walls. Results should be reported separately for each distribution shift rather than only through a random in-distribution split.

(b) **External simulations from previous work.** Evaluate the trained model, or a clearly described adapted version, on an independently generated coronary-hemodynamics dataset from prior work. An external dataset would test robustness to differences in geometry preprocessing, meshing, CFD implementation, numerical solver, and boundary-condition selection. Even if direct zero-shot evaluation is not possible because the input or output formats differ, the authors could evaluate transfer with limited fine-tuning and compare it with training from scratch.

(c) **Robustness to noisy and imperfect inputs.** The current model receives clean centerlines, radii, and flow conditions extracted from simulations, whereas real imaging and measurement pipelines introduce uncertainty. The authors should perturb the inputs using several plausible noise models and magnitudes, ideally informed by reported errors in vessel segmentation, centerline extraction, radius estimation, and flow measurement. At minimum, this could include coordinate noise, radius noise, missing or displaced centerline points, and noise in inlet flow rate. It would be useful to compare: (i) a model trained on clean data and tested on noisy data, and (ii) a model trained with noise augmentation and tested under the same noise conditions. Testing multiple noise types and levels would clarify both robustness and whether noise-aware training can recover performance.

Without such experiments, the evidence supports strong performance within the particular synthetic distribution used in the paper, but does not yet support broader claims about robustness or generalization.

**4. Failure-case examples and analysis would strengthen the empirical evaluation.** The paper reports aggregate errors and qualitative examples, but it would be useful to include a more systematic analysis of failure cases. For example, the authors could show examples where the method fails most strongly, analyze whether failures occur near severe stenoses, branches, high-curvature regions, or unusual flow conditions, and compare these failure modes to the baselines. This would make the strengths and limitations of the method clearer.

**5. The architecture overview figure could be improved.** Figure 1 gives a high-level view of the model, but it is still difficult to understand the full pipeline, especially how centerline features influence predictions at arbitrary wall query points, how the RBF kernels are parameterized, and how the different centerline contributions are aggregated. A clearer figure showing the mapping from centerline points to anisotropic kernels and then from kernels to wall-query predictions would make the method much easier to understand.

**Audience:**

Yes

**Audience Explanation:**

The paper is likely to interest a specialized but clear subset of the TMLR audience, particularly researchers working on scientific machine learning, neural operators, geometric deep learning, continuous field reconstruction, and learned surrogates for PDE-based simulations. Although coronary hemodynamics is a relatively narrow application domain, multiple recent works have developed machine-learning methods for this specific task, indicating that it is an active area within scientific and medical machine learning.
The paper also addresses a broader machine-learning question: how to exploit the intrinsic low-dimensional structure of tubular geometries to predict continuous fields efficiently. Representing a three-dimensional vessel through its one-dimensional centerline and using anisotropic RBF kernels for continuous decoding provides an interesting alternative to methods operating directly on dense surface or volumetric discretizations. The experiments do not establish generalization beyond coronary arteries, so the relevance is mainly to a specialized audience rather than to TMLR readers broadly.

**Claims And Evidence:**

No

**Claims Explanation:**

The results are convincing within the authors’ own simulation setup, where the method consistently outperforms the reported baselines and shows a favorable accuracy-efficiency tradeoff. However, I do not think the evidence fully supports the broader claims.
First, the paper frames the method as fast, non-invasive coronary hemodynamics prediction, but all evaluation targets come from synthetic CFD simulations. There is no validation against patient measurements such as invasive FFR, measured pressure, or clinically derived WSS. Thus, the evidence supports agreement with simulated outputs, not clinical accuracy. Second, the generalization evidence is limited. Train and test cases appear to come from the same geometry-generation procedure, OpenFOAM setup, physical assumptions, and parameter ranges. The paper does not test meaningful distribution shifts, external simulation datasets, different boundary conditions or physical assumptions, unseen parameter ranges, or noisy/imperfect inputs. Overall, the paper shows strong performance on the proposed datasets, but additional validation are needed.

**Requested Changes:**

**1. Clarify the motivation for fast prediction.**
The paper should more concretely explain which clinical or research workflows require faster-than-CFD prediction, what time scale is needed, and how common these workflows are. This would strengthen the motivation but is not critical by itself.

**2. Calibrate the clinical and “non-invasive” claims.**
The authors should either add validation against real patient measurements, such as invasive FFR or measured pressure, if available, or narrow the claims to simulated coronary hemodynamics. The current evidence supports agreement with CFD simulations, not a validated non-invasive clinical tool.

**3. Add stronger generalization experiments beyond the current simulation pipeline.**
The authors should evaluate distribution shifts, such as held-out ranges of stenosis severity/location, flow rate, vessel radius, curvature, branching complexity, or boundary conditions. Ideally, they should also test different physical assumptions.

**4. Evaluate on external simulations or prior datasets.**
This would strongly improve the paper and may be important for establishing robustness. Testing on independently generated coronary-hemodynamics simulations from previous work would help determine whether the method generalizes beyond the authors’ preprocessing, meshing, solver, and boundary-condition choices.

**5. Test robustness to noisy and imperfect inputs.**
This would substantially strengthen the paper. Since real pipelines introduce uncertainty in centerline extraction, radius estimation, flow measurement, and segmentation, the authors should evaluate coordinate noise, radius noise, missing/displaced centerline points, and flow-rate noise. It would be also useful to compare a clean-trained model with a model trained using noise augmentation.

**6. Add failure-case analysis.**
This would strengthen the empirical section. The authors should show cases with the largest errors and analyze whether failures occur near severe stenoses, branches, high-curvature regions, or unusual flow conditions.

**7. Improve the architecture overview figure.**
This is not critical but would improve clarity. Figure 1 should more clearly show how centerline points produce anisotropic kernels, how these kernels influence wall-query points, and how their contributions are aggregated into pressure and WSS predictions.

---

> ### Author Response · Authors · 2026-07-03
>
> We thank the reviewer for their careful reading and constructive suggestions, which have meaningfully improved the paper. We address each point below.
>
>
> **1. "The motivation for the practical need of the method should be clarified."**
>
> We appreciate this suggestion. We now describe three concrete scenarios requiring faster-than-CFD inference: point-of-care FFR estimation during imaging, interactive virtual stent placement, and cohort-scale hemodynamic phenotyping (Section 1, citing [1]).
>
> **2. "The clinical and 'non-invasive' framing is stronger than the current validation supports."**
>
> We agree this was overclaimed. We have rewritten the abstract to "learned surrogate for fast prediction of CFD-simulated coronary hemodynamics" and added an explicit disclaimer: the model "serves as a fast surrogate for the simulation pipeline rather than a replacement for clinical measurement".
>
> **3a. "Out-of-distribution simulation parameters... hold out ranges of stenosis severity, location, inlet flow rate..."**
>
> This is a fair concern. We added: (i) a patient-disjoint evaluation (Table 3) where no anatomy appears in more than one split. Our model still outperforms all baselines by 22%; (ii) stratified analysis by flow rate regime, flow rate tercile, vessel type, and branching complexity (Figure 5). True held-out parameter ranges (e.g., entirely unseen stenosis severities) require generating new CFD data (~weeks of compute); we acknowledge this limitation and note it as future work.
>
> **3b. "External simulations from previous work." and  "Evaluate on external simulations or prior datasets."**
>
> We agree this would strengthen the paper. We investigated the publicly available dataset of [2] (~4,000 coronary geometries with OpenFOAM CFD). However, direct evaluation is blocked by two representation gaps: (i) no centerline annotations, (ii) unrecorded inlet flow rates. We document these in Appendix A.15 and leave bridging them to future work.
>
> **3c. "Robustness to noisy and imperfect inputs." and "Test robustness to noisy and imperfect inputs."**
>
> We appreciate this suggestion. We added Figure 6 evaluating inference-time sensitivity to: Gaussian coordinate noise on query points, Gaussian noise on inlet flow rate, and random point dropout. Our model maintains the lowest absolute error across all noise levels. We note in the text that this robustness is partly architectural. Noise-augmented training (comparing clean-trained vs. augmented models) requires retraining all models and is noted as future work.
>
> **4. "Failure-case examples and analysis would strengthen the empirical evaluation."**
>
> Thank you for this suggestion. We added Appendix A.14 with: error distributions, geometry correlates (branches r=0.26, minimum radius r=−0.35), cross-seed consistency (r=0.88), cross-model correlation, and visualization of the 6 highest-error cases (Figure 12). Errors concentrate at stenosis sites, branch points, and distal branches of complex LCA geometries.
>
> **5. "The architecture overview figure could be improved."**
>
> Agreed. We added annotations to Figure 1 clarifying the one-kernel-per-point structure and showing explicit notation (f ∈ {P, WSS}) in the decoder, as well as arrows indicating how centerline coordinates x_{c,i} feed into the RBF decoder in addition to encoder. We are happy to further revise if specific layout suggestions are provided.
>
>
> [1] Corral-Acero et al. "The 'digital twin' to enable the vision of precision cardiology". European Heart Journal, 2020.
>
> [2] Suk et al. "Mesh convolutional neural networks for wall shear stress estimation in 3D artery models". STACOM@MICCAI, 2021.